# EnrichIndex: Using LLMs to Enrich Retrieval Indices Offline

**Peter Baile Chen**[1]    **Tomer Wolfson**[2]    **Michael Cafarella**[1]    **Dan Roth**[2,3]
[1]MIT    [2]University of Pennsylvania    [3]Oracle AI
Correspondence: peterbc@mit.edu

## Abstract

Existing information retrieval systems excel in cases where the language of target documents closely matches that of the user query. However, real-world retrieval systems are often required to *implicitly reason* whether a document is relevant. For example, when retrieving technical texts or tables, their relevance to the user query may be implied through a particular jargon or structure, rather than explicitly expressed in their content. Large language models (LLMs) hold great potential in identifying such implied relevance by leveraging their reasoning skills. Nevertheless, current LLM-augmented retrieval is hindered by high latency and computation cost, as the LLM typically computes the query-document relevance *online*, for every query anew. To tackle this issue we introduce EnrichIndex, a retrieval approach which instead uses the LLM *offline* to build semantically-enriched retrieval indices, by performing a single pass over all documents in the retrieval corpus once during ingestion time. Furthermore, the semantically-enriched indices can complement existing online retrieval approaches, boosting the performance of LLM re-rankers. We evaluated EnrichIndex on five retrieval tasks, involving passages and tables, and found that it outperforms strong online LLM-based retrieval systems, with an average improvement of 11.7 points in recall @ 10 and 10.6 points in NDCG @ 10 compared to strong baselines. In terms of online calls to the LLM, it processes 293.3 times fewer tokens which greatly reduces the online latency and cost. Overall, EnrichIndex is an effective way to build better retrieval indices offline by leveraging the strong reasoning skills of LLMs[1].

## 1 Introduction

Information retrieval plays a crucial role in various knowledge-intensive tasks, including open-domain question answering and fact-checking. While recent advancements in dense retrievers (Yu et al., 2024; Li & Li, 2023; Zhang et al., 2024) have achieved strong results on retrieval leaderboards like MTEB (Muennighoff et al., 2022), retrievers continue to struggle with more complex tasks that involve technical or domain-specific documents as well as tables (Sen et al., 2020; Chen et al., 2024a; Lei et al., 2024). These shortcomings are especially true when the relation between the target documents to the user query is not explicitly stated and needs to be implicitly inferred given the document contents (Su et al., 2024).

As such, researchers have sought to leverage the reasoning capabilities of LLMs to better rank the query-document relevance in challenging retrieval tasks. Retrieval re-ranking approaches (Glass et al., 2022; Rathee et al., 2025) typically rely on *online* LLM computations to match the user query with the most relevant documents, these may include query decomposition and expansion (Wolfson et al., 2020; Yoran et al., 2023; Chen et al., 2024b; 2025), as well as performing document expansion with respect to the user query (Niu et al., 2024). While these methods achieve significant improvements, they possess two key limitations: First, retrieval re-rankers require real-time processing for each new query, resulting in high latency and online cost due to repeated query and document expansion. Second, as ranking all documents online using LLMs is costly, these methods typically

---

[1]Data and code are available at https://peterbaile.github.io/enrichindex/.

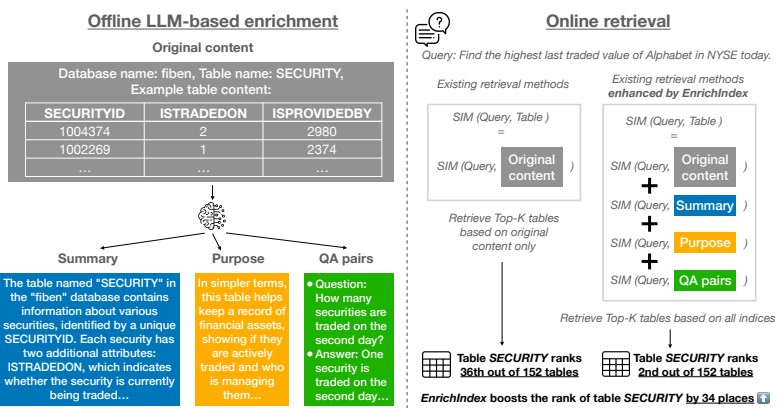

Figure 1: EnrichIndex leverages LLMs *offline* to enrich each object, creating multiple semantically-enhanced indices. During online retrieval, it computes object relevance by calculating a weighted sum of similarities between the user query across all enriched indices: the original table, its summary, purpose and QA pairs. See Appendix A for an additional example of enriching a free-form document, whose retrieval requires implicit reasoning.

rely on a lightweight *stage-one retriever* to first retrieve a moderate set of documents, which are then ranked by the LLM. As a result, the overall performance of the LLM re-ranker is inherently limited by the performance of the stage-one retriever.

To address these limitations we introduce EnrichIndex, an *offline* approach which uses LLM reasoning to improve stage-one retrieval by generating new *semantically-enriched retrieval indices*: enriching each object[2] in the corpus with multiple representations. EnrichIndex seamlessly integrates with existing stage-one retrieval methods (BM25, dense retrievers, or hybrid approaches) by computing a weighted similarity score between the user query and the enriched representations of the object, for each object in the corpus. The semantically-enriched indices help boost stage-one retrieval, leading to better performance of the downstream LLM re-rankers. Moreover, as EnrichIndex constructs its enriched indices once at an offline stage, it performs only a single object enrichment pass, which significantly reduces future computation costs compared to methods that perform (query-dependent) document expansion online.

We evaluated EnrichIndex on five retrieval datasets for documents and tables. When integrated with four different stage-one retrieval models, EnrichIndex leads to average gains of 4.75 points in recall @10 and 4.5 points in NDCG @10 with minimal online overhead. When compared to a state-of-the-art retrieve and re-rank system (Niu et al., 2024), we observe that EnrichIndex-enhanced stage-one retrievers boost the recall @10 and NDCG @10 scores by 11.7 and 10.6 points respectively, while enabling the LLM to process 293.3 times fewer tokens which significantly reduces its online latency and cost.

In summary, our contributions are as follows: (1) We introduce EnrichIndex, which leverages LLMs offline to enrich objects, significantly reducing high online overheads. (2) Through experiments on both document and table datasets, we demonstrate how EnrichIndex significantly improves retrieval performance while lowering online costs compared to state-of-the-art online re-rankers. (3) We provide a detailed analysis into the effectiveness of offline enrichment and the significance of each enrichment type in different settings.

## 2 Enriching retrieval indices offline

We focus on the standard information retrieval setting where given: (1) a document corpus or database and (2) a user query, the retriever system outputs a ranked list of documents

---

[2]We use the term *object* to refer to both free-form text documents and tables.

or tables relevant to the input query. Typically, the retriever will first index all objects (documents or tables) in the corpus at an offline phase to enable an efficient query lookup during the online query stage. For example, BM25 builds an inverted index, while dense retrieval systems index object embeddings.

Our approach enhances the retrieval indexing phase by enriching each object with three additional representations: (1) its purpose, (2) a summary, and (3) question-answer pairs derived from its content. All three of these enriched representations are then indexed alongside the original object contents.

## 2.1 Object enrichment

Consider the example in Figure 1, which features a domain-specific table on security trading, with columns described using a specific technical jargon, and a user query (top right) on the *latest* traded stock value. The relevant information requires retrieving the SECURITY table from the underlying database. However, none of the table columns appear in the query and the domain-specific naming convention also leads to a low semantic similarity with the user query. This in turn, causes top-performing dense retrievers such as GTE (Zhang et al., 2024) to rank the target table only at the 36th place. Next, we illustrate how our enriched indices help address this limitation, ultimately boosting the SECURITY table up to the 2nd spot. We introduce three enrichment methods to complement the original object content:

**Summary.** As previously discussed, retrieval methods face challenges in retrieving: (1) objects containing technical or domain-specific content and (2) objects other than free-form text. To address these issues, we propose enriching the original object with a *summary* to aid the retriever in better understanding the object contents. The summary serves as a condensed and paraphrased text describing the original content, using less technical jargon and potentially removing distracting sentences. In Figure 1, the summary explicitly states that *"security is currently being traded,"* emphasizing the concept of trading. This makes the table more likely to align with the user query, which refers to the *"highest traded value"*.

**Purpose.** While existing retrieval methods mainly rely on keyword match or semantic similarity between the query and the object, these approaches are largely ineffective when the relevance of an object requires implicit reasoning (Su et al., 2024). To address this limitation, we enrich the original object with its *purpose*. Unlike a summary, which is a re-worded and condensed version of the original content, the purpose should capture the implicit meaning and potential uses of the object. It enables the retriever to better bridge the gap between the wording of the user query and the implied purpose of objects in the retrieval corpus. In our example, the *purpose* of the target table emphasizes that it serves as *"a record of financial assets"* and describes the status of active trading, making it much more aligned semantically with the user query on the traded value of a particular stock.

**QA pairs.** Retrieval systems typically measure the similarity between the user query and an object to *indirectly* estimate the likelihood that its contents are relevant to the query. Instead, we go the other route by representing an object using a set of generated QA pairs. Comparing the user queries to sets of questions and answers should aid retrievers to more directly estimate the object relevance, providing a stronger retrieval signal to identify those that are most likely to contain the target answers. In our experiments, each object was enriched with 20 question-answer pairs. The example QA pair in Figure 1 implies that the table can be used to answer questions about trading, making it much more likely to align with the user query's mention of *"last traded value"*.

## 2.2 Online retrieval

Following the enrichment phase, each enriched object is associated with a new retrieval index, one for each representation. Overall, we have four indices for the object: one for the original content, one for the summary, one for the purpose, and one for the QA pairs.

Given an input retrieval query online, $q$, we compute its similarity with the original object content $o_b$ as well as its similarity with the object purpose $o_p$, summary $o_s$, and example QA pairs $o_{qa}$. Let $S$ represent the similarity function of the underlying retrieval method. The joint query-object similarity between $q$ and $o$ is then computed as a weighted sum of the similarity scores between $q$ and each representation:

$$S(q, o) = \alpha_1 S(q, o_b) + \alpha_2 S(q, o_p) + \alpha_3 S(q, o_s) + \alpha_4 S(q, o_{qa}) \tag{1}$$

The top-$k$ objects are then selected based on their overall similarity score $S(q, o)$ and returned as the final retrieval results. As shown in Figure 1, GTE with EnrichIndex elevates the table SECURITY by 34 positions, moving it from rank 36 to rank 2.

## 3 Experiments

### 3.1 Datasets

We evaluated our approach and baselines on complex open-domain retrieval tasks, on documents and on tables, that capture the more challenging retrieval aspects discussed in Section 1. We considered two types of tasks, *implicit retrieval* and *table retrieval*. The implicit retrieval benchmark is the Bright dataset (Su et al., 2024), which includes technical documents from domains such as StackExchange discussions, coding, and theorems, requiring implicit reasoning between the question and documents. To measure table retrieval, we evaluated on three datasets, Spider2 (Lei et al., 2024), Beaver (Chen et al., 2024a), and Fiben (Sen et al., 2020). These tasks involve retrieving the relevant tables given user questions over complex databases containing large tables and domain-specific knowledge. Since some table retrieval datasets contain fewer than 10 tables for some databases, the original setup makes retrieval too easy (e.g., retrieving 10 tables from a database with 10 tables results in a recall of 100). To address this, we combined all databases into a single unified central database, increasing the corpus size and making the retrieval task more challenging. Lastly, to validate that our approach also excels on simpler retrieval tasks, we included NQ (Kwiatkowski et al., 2019), a standard document retrieval task without the aforementioned challenges. Details about dataset statistics can be found in Appendix C.

### 3.2 Measuring retrieval performance and efficiency

Following standard retrieval metrics, we evaluated retrieval performance using precision, recall, F1 and NDCG @$k$.

Additionally, we evaluated efficiency from two perspectives: latency and cost, by analyzing the number of input and output tokens used by LLMs online. On the same hardware, processing fewer input tokens and generating fewer output tokens reduces the workload for LLMs, leading to lower latency. Likewise, fewer processed tokens decrease computational demands, lowering costs. Since both latency and cost are directly influenced by token usage, we used token count as a key metric for measuring efficiency.

We tracked LLM usage only during the online phase, not for offline enrichment. In scenarios where the object corpus remains largely static, EnrichIndex conducts enrichment once during the offline stage, allowing the enriched representations to be reused across all queries. As query volume increases, the amortized cost of enrichment can become very small.

### 3.3 Baselines and setup

Since our method aims to enhance existing retrieval techniques by incorporating enriched object representations, we compared it against standard retrieval methods that rely solely on the original object content. For table retrieval datasets, each table was serialized to include its table name, columns, and a randomly selected sample of rows (full details in Appendix D) as used by past works (Chen et al., 2025; Lei et al., 2024). We evaluated three common retrieval frameworks: (1) BM25, (2) dense retrievers, and (3) a hybrid approach that combines both. The retrieval results can then be further refined using LLMs as re-rankers (Niu et al., 2024). To maintain clarity within the retrieve-and-re-rank framework, we refer to

| | k = 10 | | | | | | | | | | | | k = 100 | | | | | | | | | | | |
|---|---|---|---|---|---|---|---|---|---|---|---|---|---|---|---|---|---|---|---|---|---|---|---|---|---|
| | StackEx. | | Coding | | Thm. | | Avg. | | Improvement | | | | StackEx. | | Coding | | Thm. | | Avg. | | Improvement | | | |
| | R | N | R | N | R | N | R | N | R(pt.) | R(%) | N(pt.) | N(%) | R | N | R | N | R | N | R | N | R(pt.) | R(%) | N(pt.) | N(%) |
| *Original question* | | | | | | | | | | | | | | | | | | | | | | | | |
| BM25 | 23.4 | 18.9 | 17.1 | 10.4 | 5.5 | 3.8 | 17.3 | 13.2 | - | - | - | - | 48.1 | 25.8 | 31.7 | 15.0 | 14.1 | 6.0 | 35.7 | 18.4 | - | - | - | - |
| BM25$_E$ | 21.7 | 17.6 | 14.8 | 9.8 | 5.8 | 4.0 | 16.0 | 12.4 | -1.3 | -7.5 | -0.8 | -6.1 | 49.7 | 25.6 | 31.8 | 14.8 | 14.6 | 6.2 | 36.8 | 18.3 | +1.1 | +3.1 | -0.1 | -0.5 |
| UAE | 19.6 | 16.5 | 15.4 | 10.5 | 8.3 | 5.5 | 15.7 | 12.4 | - | - | - | - | 47.4 | 24.2 | 32.7 | 15.7 | 19.7 | 8.4 | 37.1 | 18.3 | - | - | - | - |
| UAE$_E$ | 22.9 | 19.1 | 18.5 | 14.1 | 8.8 | 6.2 | 18.2 | 14.6 | +2.5 | +15.9 | +2.2 | +17.7 | 52.1 | 27.3 | 35.3 | 18.9 | 23.7 | 9.8 | 41.2 | 20.9 | +4.1 | +11.1 | +2.6 | +14.2 |
| GTE | 18.7 | 15.0 | 13.9 | 11.9 | 9.2 | 6.1 | 15.2 | 11.9 | - | - | - | - | 47.8 | 23.1 | 37.6 | 18.9 | 24.3 | 10.0 | 39.4 | 18.7 | - | - | - | - |
| GTE$_E$ | 20.9 | 16.6 | 18.6 | 15.0 | 13.4 | 9.3 | 18.4 | 14.3 | +3.2 | +21.2 | +2.4 | +20.2 | 52.3 | 25.4 | 43.8 | 22.9 | 27.4 | 12.7 | 43.9 | 21.5 | +4.5 | +11.4 | +2.8 | +15.0 |
| Snowflake | 22.3 | 18.9 | 15.6 | 10.4 | 10.4 | 6.8 | 17.8 | 14.0 | - | - | - | - | 46.7 | 25.6 | 34.7 | 15.9 | 24.2 | 10.2 | 38.3 | 19.6 | - | - | - | - |
| Snowflake$_E$ | 25.7 | 21.6 | 20.1 | 16.9 | 12.1 | 7.7 | 20.9 | 16.9 | +3.1 | +17.4 | +2.9 | +20.7 | 56.0 | 30.2 | 46.4 | 24.4 | 27.8 | 11.7 | 46.4 | 24.0 | +8.1 | +21.1 | +4.4 | +22.4 |
| BM25+UAE | 23.9 | 20.1 | 18.0 | 12.5 | 8.9 | 5.7 | 18.7 | 14.7 | - | - | - | - | 53.6 | 28.5 | 37.8 | 18.6 | 22.8 | 9.3 | 42.2 | 21.4 | - | - | - | - |
| (BM25+UAE)$_E$ | 25.4 | 21.4 | 17.4 | 13.4 | 9.7 | 6.4 | 19.6 | 15.8 | +0.9 | +4.8 | +1.1 | +7.5 | 58.4 | 30.6 | 38.8 | 19.6 | 24.5 | 9.9 | 45.4 | 22.8 | +3.2 | +7.6 | +1.4 | +6.5 |
| BM25+GTE | 25.6 | 21.0 | 18.7 | 15.5 | 11.6 | 7.3 | 20.5 | 16.2 | - | - | - | - | 54.7 | 29.2 | 44.6 | 23.3 | 26.7 | 11.1 | 45.1 | 23.1 | - | - | - | - |
| (BM25+GTE)$_E$ | 26.8 | 22.1 | 20.7 | 16.8 | 13.2 | 9.1 | 21.9 | 17.5 | +1.4 | +6.8 | +1.3 | +8.0 | 57.4 | 30.5 | 47.8 | 24.8 | 29.8 | 13.3 | 48.0 | 24.7 | +2.9 | +6.4 | +1.6 | +6.9 |
| BM25+Snow. | 27.2 | 22.2 | 17.9 | 12.1 | 10.4 | 6.9 | 20.8 | 16.1 | - | - | - | - | 54.7 | 30.1 | 40.6 | 18.8 | 24.5 | 10.5 | 43.8 | 22.6 | - | - | - | - |
| (BM25+Snow.)$_E$ | 27.2 | 21.9 | 20.8 | 17.3 | 12.0 | 7.9 | 21.9 | 17.2 | +1.1 | +5.3 | +1.1 | +6.8 | 59.6 | 31.3 | 47.4 | 24.8 | 28.0 | 11.9 | 48.7 | 24.8 | +4.9 | +11.2 | +2.2 | +9.7 |
| Average | 23.0 | 18.9 | 16.7 | 11.9 | 9.2 | 6.0 | 18.0 | 14.1 | - | - | - | - | 50.4 | 26.7 | 37.1 | 18.0 | 22.3 | 9.4 | 40.2 | 20.3 | - | - | - | - |
| Average$_E$ | 24.4 | 20.0 | 18.7 | 14.8 | 10.7 | 7.2 | 19.6 | 15.5 | +1.6 | +8.9 | +1.4 | +9.9 | 55.1 | 28.7 | 41.6 | 21.5 | 25.1 | 10.8 | 44.3 | 22.4 | +4.1 | +10.2 | +2.1 | +10.3 |
| *GPT-4 generated expanded query* | | | | | | | | | | | | | | | | | | | | | | | | |
| Average | 33.5 | 29.1 | 15.5 | 12.1 | 16.3 | 11.7 | 25.4 | 21.2 | - | - | - | - | 61.7 | 37.0 | 35.3 | 18.2 | 30.8 | 15.3 | 48.3 | 27.6 | - | - | - | - |
| Average$_E$ | 34.6 | 30.1 | 17.5 | 15.6 | 19.4 | 13.8 | 27.3 | 23.0 | +1.9 | +7.5 | +1.8 | +8.5 | 64.9 | 38.5 | 42.4 | 22.8 | 35.2 | 17.8 | 52.5 | 29.9 | +4.2 | +8.7 | +2.3 | +8.3 |

Table 1: Recall (R) and NDCG (N) @*k* of stage-one retrievers on the Bright dataset. X$_E$ refers to retrieval method X augmented with EnrichIndex. Average and Average$_E$ refer to the average performance of all methods without and with EnrichIndex, respectively. Gray numbers indicate the performance of retrievers when they achieve lower performance without EnrichIndex. **Bolded** numbers indicate the performance gain (in points (pt.) and percentages (%)) of retrievers when they achieve higher performance with EnrichIndex. Refer to Appendix F for complete numbers.

retrieval using BM25, dense retrievers, or the hybrid method as *stage-one retrieval*. In our experiments, we fed re-rankers the top $k = 100$ documents on the Bright dataset and $k = 20$ tables on the table retrieval datasets. For the Bright dataset, we set $k = 100$, following the approach in Su et al. (2024); Niu et al. (2024). For the table retrieval datasets, we chose $k = 20$ since the average number of gold tables required per question is 3.54, and selecting 20 allows for a broader set of objects to be included for reranking. Details for hyperparameter tuning can be found in Appendix E.

For BM25, we used the implementation from Li et al. (2024). For dense retrieval, we selected lightweight models under 500M parameters, including UAE-Large-V1 (Li & Li, 2023), GTE-multilingual-base (Zhang et al., 2024), and Snowflake-arctic-l (Yu et al., 2024), all of which rank among the top models on the MTEB leaderboard for retrieval tasks.

We compared EnrichIndex with a strong retrieval re-ranking approach that relies on complex *online* analysis using LLM reasoning. For the challenging retrieval tasks mentioned earlier, a common strategy is to retrieve the top-*k* objects from stage-one retrieval and then analyze each object's relevance using an LLM before ranking them accordingly. To this end, we compared our method with Judgerank, a powerful retrieval re-ranking approach and the current state-of-the-art for the Bright benchmark[3]. Judgerank employs BM25 for stage-one retrieval, followed by an online query expansion and question-specific document expansion phase that is used for re-ranking the documents. The implementation details are provided in Appendix B.1. We did not evaluate Judgerank on the NQ dataset, as it is less challenging and existing stage-one retrievers already achieve high performance.

The enrichment process of our method is performed offline using GPT-4o-mini, with the specific prompts detailed in Appendix B.2.

### 3.4 Performance

**Performance of stage-one retrievers.** We begin by examining whether EnrichIndex can improve the retrieval performance of various stage-one retrieval methods. As shown in the upper sections of Tables 1, 2 and 3, EnrichIndex generally improves retrieval performance for stage-one retrieval methods. For the Bright dataset, across all of its domains, EnrichIndex enhances the recall and NDCG @10 of dense retrievers by an average of 2.93 and 2.50 points respectively. Similarly, it improves recall and NDCG @10 of hybrid approaches by an average of 1.13 and 1.17 points. When looking at the three table retrieval datasets,

---

[3]https://brightbenchmark.github.io/ as of April 2nd, 2025

| | k = 10 | | | | | | | | | | | | k = 20 | | | | | | | | | | | |
|---|---|---|---|---|---|---|---|---|---|---|---|---|---|---|---|---|---|---|---|---|---|---|---|---|---|
| | Spider2 | | Beaver | | Fiben | | Avg. | | Improvement | | | | Spider2 | | Beaver | | Fiben | | Avg. | | Improvement | | | |
| | R | N | R | N | R | N | R | N | R(pt.) | R(%) | N(pt.) | N(%) | R | N | R | N | R | N | R | N | R(pt.) | R(%) | N(pt.) | N(%) |
| *Original question* | | | | | | | | | | | | | | | | | | | | | | | | |
| BM25 | 43.3 | 32.0 | 51.7 | 43.9 | 30.6 | 29.0 | 40.6 | 34.1 | - | - | - | - | 53.0 | 35.1 | 67.7 | 50.0 | 33.8 | 30.2 | 49.5 | 37.2 | - | - | - | - |
| BM25$_E$ | 60.9 | 46.5 | 59.2 | 52.4 | 47.4 | 40.9 | 55.2 | 45.9 | +14.6 | +36.0 | +11.8 | +34.6 | 69.8 | 49.6 | 73.9 | 58.4 | 62.9 | 46.6 | 68.2 | 50.8 | +18.7 | +37.8 | +13.6 | +36.6 |
| UAE | 55.4 | 44.2 | 48.6 | 43.8 | 48.2 | 42.6 | 50.7 | 43.5 | - | - | - | - | 70.2 | 48.9 | 67.5 | 51.2 | 61.4 | 47.8 | 66.0 | 49.1 | - | - | - | - |
| UAE$_E$ | 60.2 | 47.0 | 54.8 | 49.3 | 55.3 | 52.0 | 56.8 | 49.6 | +6.1 | +12.0 | +6.1 | +14.0 | 74.2 | 51.7 | 72.4 | 56.3 | 63.3 | 55.3 | 69.5 | 54.3 | +3.5 | +5.3 | +5.2 | +10.6 |
| GTE | 53.8 | 41.9 | 54.0 | 45.9 | 54.4 | 56.9 | 54.1 | 48.8 | - | - | - | - | 62.4 | 44.7 | 69.6 | 52.2 | 63.1 | 60.4 | 64.7 | 52.9 | - | - | - | - |
| GTE$_E$ | 61.5 | 47.0 | 58.0 | 51.7 | 60.7 | 62.7 | 60.2 | 54.4 | +6.1 | +11.3 | +5.6 | +11.5 | 72.5 | 50.7 | 74.0 | 58.1 | 67.2 | 65.3 | 70.8 | 58.4 | +6.1 | +9.4 | +5.5 | +10.4 |
| Snowflake | 45.1 | 34.1 | 49.2 | 42.3 | 51.8 | 47.5 | 48.8 | 41.6 | - | - | - | - | 53.1 | 36.5 | 66.3 | 49.3 | 58.7 | 50.4 | 58.9 | 45.4 | - | - | - | - |
| Snowflake$_E$ | 62.0 | 48.1 | 58.0 | 52.4 | 53.9 | 53.8 | 57.7 | 51.5 | +8.9 | +18.2 | +9.9 | +23.8 | 71.3 | 51.0 | 74.1 | 59.0 | 59.4 | 56.0 | 67.4 | 55.1 | +8.5 | +14.4 | +9.7 | +21.4 |
| BM25+UAE | 65.2 | 50.7 | 55.3 | 48.0 | 44.8 | 37.8 | 54.5 | 45.0 | - | - | - | - | 74.2 | 53.9 | 76.2 | 56.2 | 61.5 | 44.3 | 69.8 | 50.8 | - | - | - | - |
| (BM25+UAE)$_E$ | 67.7 | 53.0 | 62.9 | 54.2 | 52.1 | 43.2 | 60.3 | 49.5 | +5.8 | +10.6 | +4.5 | +10.0 | 77.7 | 56.4 | 79.2 | 60.8 | 63.1 | 47.5 | 72.4 | 54.1 | +2.6 | +3.7 | +3.3 | +6.5 |
| BM25+GTE | 60.2 | 48.1 | 57.1 | 49.1 | 51.8 | 43.8 | 56.1 | 46.7 | - | - | - | - | 69.4 | 51.1 | 77.2 | 56.9 | 62.5 | 48.1 | 68.8 | 51.5 | - | - | - | - |
| (BM25+GTE)$_E$ | 68.1 | 52.5 | 64.7 | 56.5 | 58.6 | 48.3 | 63.5 | 51.9 | +7.4 | +13.2 | +5.2 | +11.1 | 76.1 | 55.2 | 80.7 | 62.8 | 66.9 | 51.6 | 73.8 | 55.9 | +5.0 | +7.3 | +4.4 | +8.5 |
| BM25+Snow. | 55.1 | 42.8 | 55.5 | 47.0 | 50.3 | 42.0 | 53.4 | 43.6 | - | - | - | - | 64.3 | 45.8 | 75.2 | 54.7 | 59.3 | 45.5 | 65.3 | 48.1 | - | - | - | - |
| (BM25+Snow.)$_E$ | 68.7 | 53.0 | 63.9 | 55.5 | 50.6 | 44.9 | 60.3 | 50.5 | +6.9 | +12.9 | +6.9 | +15.8 | 77.8 | 55.9 | 80.4 | 62.1 | 59.2 | 48.3 | 71.2 | 54.6 | +5.9 | +9.0 | +6.5 | +13.5 |
| Average | 54.0 | 42.0 | 53.1 | 45.7 | 47.4 | 42.8 | 51.2 | 43.3 | - | - | - | - | 63.8 | 45.1 | 71.4 | 52.9 | 57.2 | 46.7 | 63.3 | 47.9 | - | - | - | - |
| Average$_E$ | 64.1 | 49.6 | 60.2 | 53.1 | 54.1 | 49.4 | 59.1 | 50.5 | +7.9 | +15.4 | +7.2 | +16.6 | 74.2 | 52.9 | 76.4 | 59.6 | 63.1 | 52.9 | 70.5 | 54.7 | +7.2 | +11.4 | +6.8 | +14.2 |
| *GPT-4o-mini generated expanded query* | | | | | | | | | | | | | | | | | | | | | | | | |
| Average | 53.8 | 41.5 | 51.9 | 43.9 | 53.4 | 45.6 | 53.1 | 43.7 | - | - | - | - | 63.5 | 44.5 | 68.6 | 50.5 | 71.5 | 52.5 | 68.0 | 49.3 | - | - | - | - |
| Average$_E$ | 63.1 | 49.1 | 57.9 | 51.5 | 60.5 | 51.9 | 60.7 | 50.9 | +7.6 | +14.3 | +7.2 | +16.5 | 73.7 | 52.6 | 72.7 | 57.5 | 73.3 | 56.8 | 73.3 | 55.6 | +5.3 | +7.8 | +6.3 | +12.8 |

Table 2: Recall and NDCG @*k* of stage-one retrievers on the table retrieval datasets.

| | k = 10 | | | | | | k = 100 | | | | | |
|---|---|---|---|---|---|---|---|---|---|---|---|---|
| | NQ | | Improvement | | | | NQ | | Improvement | | | |
| | R | N | R(pt.) | R(%) | N(pt.) | N(%) | R | N | R(pt.) | R(%) | N(pt.) | N(%) |
| Average | 88.4 | 74.2 | - | - | - | - | 96.5 | 76.1 | - | - | - | - |
| Average$_E$ | 89.9 | 76.0 | +1.5 | +1.7 | +1.8 | +2.4 | 97.4 | 77.7 | +0.9 | +0.9 | +1.6 | +2.1 |

Table 3: Average recall and NDCG @*k* of stage-one retrievers on the NQ dataset.

EnrichIndex boosts recall and NDCG @10 for BM25 by an average of 14.6 and 11.8 points, for dense retrievers by 7.03 and 7.2 points, and for hybrid approaches by 6.7 and 5.53 points. For the NQ dataset, EnrichIndex improves recall and NDCG @10 across all methods by an average of 1.5 and 1.8 points, respectively.

When retrieving a higher number of top-*k* objects, the improvements remain substantial or even increase. On the Bright dataset, with $k = 100$, EnrichIndex enhances recall and NDCG for dense retrievers by 5.57 and 3.27 points and for hybrid approaches by 3.67 and 1.73 points. For table datasets, at $k = 20$, EnrichIndex increases recall and NDCG for BM25 by 18.7 and 13.6 points, for dense retrievers by 6.03 and 6.80 points, and for hybrid approaches by 4.5 and 4.73 points. For the NQ dataset, at $k = 100$, EnrichIndex increases average recall and NDCG across all methods by an average of 0.9 and 1.6 points, respectively. These results indicate that strong re-rankers can process a larger set of input objects, potentially achieving even greater retrieval performance at smaller $k$.

**Performance of stage-one retrievers with query expansion.** As discussed in Section 3.3, query expansion is a widely used online technique to improve retrieval performance. To explore whether our method can further enhance performance when combined with expanded queries, we replaced the original user query with an LLM-generated expanded query and used it to compute query-object similarity for top-*k* retrieval.

Examining the lower sections of Table 1 and Table 2 reveals that EnrichIndex can also enhance various stage-one retrieval methods when using LLM-generated expanded queries instead of the original user queries. On the Bright dataset, across all methods, EnrichIndex improves recall and NDCG @10 by an average of 1.9 and 1.8 points, respectively, and enhances recall and NDCG @100 by 4.2 and 2.3 points. For table retrieval datasets, across all methods, EnrichIndex increases recall and NDCG @10 by an average of 7.6 and 7.2 points, while @20, recall and NDCG improve by 5.3 and 6.3 points, respectively.

Additionally, we find that with query expansion, our method leads to the most significant improvements in the coding and theorem domains within the Bright dataset, as well as in the table retrieval datasets. Across all retrieval methods, the average recall and NDCG @10 improvements are 1.1 and 1.0 for the Stack Exchange domain, 2.0 and 3.5 for coding, 3.1 and 2.1 for theorem, and 7.6 and 7.2 for table retrieval. This may be due to the greater semantic gap between user queries and target objects. While StackExchange documents

| | StackEx. | | Coding | | Thm. | | Average | | Improvement wrt. Judgerank | | | | Spider2 | | Beaver | | Fiben | | Average | | Improvement wrt. Judgerank | | | |
|---|---|---|---|---|---|---|---|---|---|---|---|---|---|---|---|---|---|---|---|---|---|---|---|---|
| | R | N | R | N | R | N | R | N | R(pt.) | R(%) | N(pt.) | N(%) | R | N | R | N | R | N | R | N | R(pt.) | R(%) | N(pt.) | N(%) |
| BM25 | 37.7 | 32.8 | 12.2 | 9.5 | 13.7 | 9.8 | 26.4 | 22.2 | - | - | - | - | 39.1 | 28.6 | 47.0 | 37.5 | 46.2 | 34.5 | 44.0 | 33.3 | - | - | - | - |
| Judgerank[4] (BM25-J) | 39.3 | 33.9 | 13.2 | 9.9 | 15.3 | 10.9 | 27.9 | 23.1 | - | - | - | - | 40.2 | 30.3 | 47.8 | 38.8 | 46.2 | 34.7 | 44.6 | 34.3 | - | - | - | - |
| EnrichIndex | 39.7 | 34.7 | 20.2 | 17.0 | 20.7 | 14.4 | 30.9 | 25.8 | **+3.0** | **+10.8** | **+2.7** | **+11.7** | 66.4 | 51.7 | 62.1 | 55.8 | 65.7 | 51.5 | 65.0 | 52.7 | **+20.4** | **+45.7** | **+18.4** | **+53.6** |
| EnrichIndex-J | 40.0 | 35.5 | 20.3 | 17.8 | 21.2 | 14.7 | 31.2 | 26.5 | **+3.3** | **+11.8** | **+3.4** | **+14.7** | 66.9 | 52.6 | 61.8 | 55.7 | 65.6 | 51.7 | 65.0 | 53.1 | **+20.4** | **+45.7** | **+18.8** | **+54.8** |

Table 4: Retrieval performance @10 of various methods on Bright and the table retrieval datasets *with query expansion*. X-J refers to stage-one retrieval method X augmented with the Judgerank online reranker executed on Llama-3.1-8B-Instruct. EnrichIndex-enhanced stage-one retrievers outperform Judgerank even without applying Judgerank online reranker. Incorporating Judgerank reranker on top of EnrichIndex-enhanced stage-one retrievers further improves performance.

| | StackExchange | Coding | Theorems | Average | Spider2 | Beaver | Fiben | Average |
|---|---|---|---|---|---|---|---|---|
| #Input tokens | 1072.9x | 872.4x | 1903.5x | 1158.5x | 4016.2x | 744.9x | 543.7x | 1998.8x |
| #Output tokens | 56.0x | 53.4x | 59.1x | 56.5x | 9.2x | 10.0x | 7.7x | 8.9x |
| #Total tokens | 348.4x | 327.7x | 374.8x | 351.0x | 530.8x | 99.3x | 56.4x | 235.5x |

Table 5: Reduction factor in the number of input, output, and total tokens used by LLMs when using EnrichIndex-enhanced stage-one retrievers instead of Judgerank. Latency and cost are proportional to token counts, so larger reductions are better. Refer to Appendix G for the absolute token counts.

are primarily written in free-form text, documents in the coding and theorem domains often contain programming code or mathematical equations, and tables are structured using column names and rows. Moreover, the further improvement achieved by our method beyond query expansion suggests that bridging this gap requires not only refining queries but also enhancing object representation, highlighting the necessity of our method.

**Re-ranker performance.** We investigate whether stage-one retrievers enhanced by EnrichIndex can further complement existing online rerankers to enhance overall retrieval performance. Tables 1 and 2 show that the most effective retrieval method enhanced by EnrichIndex is the hybrid approach combining the Snowflake dense retriever with BM25 on the Bright dataset and the GTE dense retriever with BM25 on the table datasets. Therefore, we selected these as the stage-one retrievers.

Table 4 demonstrates that our method can enhance existing online re-rankers, such as Judgerank, to achieve higher scores. As mentioned in Section 3.3, the original implementation of Judgerank uses BM25 as the stage-one retriever. Replacing BM25 with the stage-one retriever with EnrichIndex improves recall and NDCG @10 by 3.3 and 3.4 points on the Bright dataset and by 20.4 and 18.8 points on the table datasets. These results highlight the importance of high-quality stage-one retrieval, as re-ranker performance is inherently limited by the effectiveness of the initial retrieval stage. Improving stage-one retrieval is therefore likely to result in better overall re-ranker performance.

**Efficiency.** Finally, we compare stage-one retrievers enhanced by EnrichIndex with Judgerank in terms of both performance and efficiency, including latency and cost, using the same evaluation settings as for reranker performance. As shown in Tables 4 and 5, EnrichIndex-enhanced stage-one retrievers with online query expansion alone, *without a Judgerank-style online re-ranker*, significantly outperforms Judgerank while achieving substantially lower latency and cost, thanks to the drastically reduced number of tokens processed.

On the Bright dataset, EnrichIndex surpasses Judgerank by 3.0 and 2.7 points in recall and NDCG @10, respectively, while processing 1158.5 times fewer input tokens and 56.5 times fewer output tokens, resulting in significant latency and cost reduction. For the table

---

[4]Since the Judgerank code was not released, we made our best effort to replicate it. Due to computational resource constraints, running Judgerank on larger models like Llama-3.1-70B-Instruct is extremely time-consuming, so we limited our testing to the 8B version. Additionally, the latency and cost of running larger models would increase significantly.

|                     | Bright  | Spider2 | Beaver  | Fiben   |
|---------------------|---------|---------|---------|---------|
| Without EnrichIndex | 0.0550  | 0.1206  | 0.0915  | 0.0575  |
| With EnrichIndex    | 0.0579  | 0.1514  | 0.1069  | 0.0802  |
| % increase          | +5.3%   | +58.1%  | +29.1%  | +42.9%  |

Table 6: The Wasserstein distance between distributions of cosine similarity for (query, non-gold objects) and (query, gold objects), computed with and without EnrichIndex.

datasets, EnrichIndex outperforms Judgerank by 20.4 and 18.4 points in recall and NDCG @10, while reducing input token processing by 1998.8 times and output token processing by 8.9 times, again leading to significant latency and cost reduction. This substantial decrease in token usage stems from EnrichIndex eliminating the need for costly online query-dependent object expansion for each retrieved object in the first-stage retrieval.

This clearly highlights the significance of an improved stage-one retrieval system. A stronger stage-one retriever, when combined with lightweight online techniques like query expansion, can already surpass a system that relies on a less powerful stage-one retriever but uses a powerful online LLM-based reranker. This not only reduces costs but also lowers latency. Furthermore, as previously discussed, a better stage-one retriever can enhance online rerankers, leading to even higher performance. Therefore, our approach provides users with the flexibility to pair EnrichIndex-enhanced stage-one retrievers with either efficient, lightweight online techniques or more costly but powerful online rerankers, depending on their performance-efficiency tradeoffs.

## 4 Analysis

### 4.1 Distribution distance with and without enrichment

Under our retrieval setting, gold objects are those that contain the correct answers for each user query. Ideally, the enrichment introduced by EnrichIndex should assist retrievers in better differentiating gold objects from non-gold ones, enabling them to rank gold objects higher and improve retrieval performance. Therefore, we examine whether object enrichment indeed provides a stronger signal to retrievers in distinguishing between gold and non-gold objects. We do this by examining the distribution shift in query-object cosine similarity before and after enrichment between (query, gold object) pairs and (query, non-gold object) pairs for all queries. In this analysis, we focused on the results of best-performing dense retrievers: Snowflake for Bright and GTE for the table retrieval datasets. Specifically, we quantify the distribution shift using the Wasserstein distance[5]. Table 6 shows that the distance between gold and non-gold similarities increases significantly after object enrichment. This indicates that EnrichIndex enhances the retriever's ability to differentiate between relevant and non-relevant objects by increasing their separation. As a result, gold objects are ranked even higher than non-gold objects after enrichment, ultimately improving retrieval performance.

### 4.2 Performance breakdown by enrichment types

As described in Section 2.1, there are three types of object enrichment: purpose, summary, and QA pairs. We examine the significance of each enrichment type. In particular, we analyze the average performance of all dense retrievers in the following case: no enrichment; one type of enrichment; two types of enrichment; all three types being used. As seen in Figure 2, each enrichment positively contributes to the retrieval performance. Moreover, we observe that having all enrichment types provides the highest performance gain, higher compared to using only two types of enrichment, which is in turn higher than the performance gain using only one type of enrichment.

---

[5]scipy.stats.wasserstein_distance

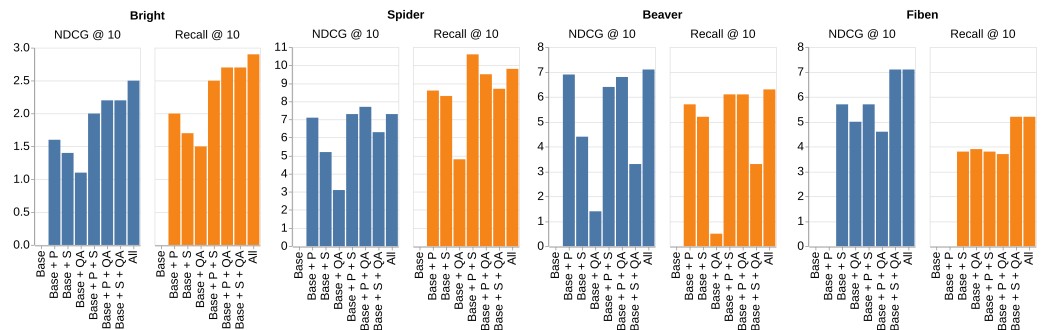

Figure 2: Absolute improvement (in points) of retrieval performance with different types of enrichment relative to using only the original object content. Base refers to original object content, P refers to purpose, S refers to summary, and QA refers to QA pairs.

| | StackEx. | | Coding | | Thm. | | Avg. | | Improvement | | | | Spider2 | | Beaver | | Fiben | | Avg. | | Improvement | | | |
|---|---|---|---|---|---|---|---|---|---|---|---|---|---|---|---|---|---|---|---|---|---|---|---|---|
| | R | N | R | N | R | N | R | N | R(pt.) | R(%) | N(pt.) | N(%) | R | N | R | N | R | N | R | N | R(pt.) | R(%) | N(pt.) | N(%) |
| *Original question* | | | | | | | | | | | | | | | | | | | | | | | | |
| Average | 22.6 | 18.7 | 16.5 | 11.8 | 9.3 | 6.2 | 17.8 | 13.9 | - | - | - | - | 53.8 | 41.5 | 53.5 | 45.8 | 47.1 | 42.6 | 51.1 | 43.1 | - | - | - | - |
| Average$_E$ | 24.1 | 20.0 | 17.8 | 13.5 | 10.8 | 7.3 | 19.3 | 15.3 | +1.5 | +8.4 | +1.4 | +10.1 | 63.7 | 50.1 | 58.2 | 51.0 | 52.2 | 47.6 | 57.7 | 49.3 | +6.6 | +12.9 | +6.2 | +14.4 |
| *GPT generated expanded query* | | | | | | | | | | | | | | | | | | | | | | | | |
| Average | 33.5 | 29.3 | 15.6 | 12.2 | 16.8 | 11.9 | 25.6 | 21.4 | - | - | - | - | 52.2 | 40.0 | 51.9 | 43.7 | 53.2 | 45.1 | 52.5 | 43.0 | - | - | - | - |
| Average$_E$ | 34.7 | 30.3 | 17.2 | 14.3 | 19.5 | 14.0 | 27.3 | 22.9 | +1.7 | +6.6 | +1.5 | +7.0 | 62.8 | 48.9 | 56.7 | 50.4 | 57.6 | 50.0 | 59.1 | 49.8 | +6.6 | +12.6 | +6.8 | +15.8 |

Table 7: Recall and NDCG @10 of stage-one retrievers on Bright and table retrieval datasets.

Regarding the relative importance of each enrichment, we observe that on the Bright dataset, which requires implicit reasoning, object-purpose contributes the most to the overall retrieval performance. Adding the purpose improves recall @ 10 and NDCG @ 10 by 2.0 and 1.6 points, respectively, compared to 1.7 and 1.4 points for object-summary, and 1.5 and 1.1 points for adding QA pairs. This may be due to the purpose enrichment's text highlighting the potential uses of the original object content, making it semantically closer to the user query. Additionally, QA pairs provide a greater performance boost than the object-summary, contributing an additional 0.7 and 0.6 points for recall and NDCG @ 10. Finally, adding summary on top of both purpose and QA pairs yields a further improvement of 0.2 and 0.3 points for recall and NDCG @ 10, respectively.

For table retrieval datasets, the object-summary provides the greatest improvement in overall retrieval performance when only a single enrichment type is used. Across all datasets, adding summary increases recall @ 10 by 5.77 points and NDCG @ 10 by 5.10 points, outperforming purpose, which improves recall @ 10 and NDCG @ 10 by 4.77 and 4.67 points, respectively, and QA pairs, which contribute 3.07 and 3.17 points. This advantage likely arises because the summary translates the tabular data into a free-form text format on which retrievers are typically optimized. On top of object-summary, adding the purpose has a greater performance boost than QA pairs, increasing recall @ 10 by 1.06 points and NDCG @ 10 by 1.37 points. Finally, incorporating QA pairs alongside both summary and purpose provides an additional improvement of 0.27 points for recall @ 10 and 0.7 points for NDCG @ 10.

Overall, based on our analysis, we conclude that in domains that require more implicit reasoning the *object-purpose* provides the greatest benefit, followed by *QA pairs*, and then the *object-summary*. In contrast, in domains where understanding contents from different modalities plays a key role, *summary* is the most effective, followed by *purpose*, then *QA pairs*.

## 4.3 Hyperparameter analysis

The experiments in Section 3.4 was performed by tuning the weights in Equation 1 individually for each dataset. To evaluate how well these weights generalize across different tasks, we also report results using a single, shared set of weights for all datasets. As shown in

Table 7, EnrichIndex continues to provide significant performance gains, highlighting the robustness and generalizability of our method across varied workloads.

## 5 Related work

**Query expansion.**   To bridge the semantic gap between user queries and documents, prior work has explored various query expansion techniques to enhance retrieval by using the expanded objects for search (Xu & Croft, 1996; Carpineto & Romano, 2012). Recent methods for query expansion include Wang et al. (2024) that augment the original query using six additional questions based on the 5W1H framework (Who, What, When, Where, Why, How). Gao et al. (2023); Wang et al. (2023) prompt LLMs to generate hypothetical answer passages conditioned solely on the original query. Follow-up work (Sun et al., 2025; Shen et al., 2024) extends this idea by incorporating retrieved documents alongside the original query to produce more informed hypothetical answers.

These approaches are complementary to ours in that they focus on enriching queries during retrieval time, whereas our method enriches documents offline. Importantly, relying solely on online query processing can substantially increase both the QA latency and computation cost, since the expansion must be performed anew for each query. In contrast, EnrichIndex performs a one-time process of enriching documents offline once during ingestion, which leads to a significant improvement in retrieval performance.

**Document expansion.**   Other lines of work enhance retrieval by expanding the documents themselves. Doc2Query (Nogueira et al., 2019) uses fine-tuned models to generate questions that can be answered from the document, effectively enriching it with potential queries. Tan et al. (2025) leverages off-the-shelf LLMs to generate additional QA pairs and event information from documents. Sarthi et al. (2024) produces document summaries and clusters documents based on these summaries. The resulting enriched content is then added back into the corpus and used in an embedding-based retrieval system.

EnrichIndex differs from these prior methods in several ways. First, it performs enrichment across multiple modalities (including text and tables) and includes diverse types of enrichment like purpose, summaries, and synthetic QA pairs. This is done using general-purpose, off-the-shelf models rather than task-specific fine-tuned ones. Second, our approach departs from prior work in how the enriched content is used during retrieval. Rather than appending or merging new content into the original document (which alters the content or corpus), we maintain both the original corpus and index. During retrieval, we consider multiple enrichment indices and combine its scores with those from the original content using a weighted sum. This modular design enables a more flexible and extensible vector storage: each index remains independent, allowing new embeddings or enrichment types to be added without disrupting the existing setup.

## 6 Conclusion

Retrieval tasks involving technical texts and domain-specific tables often require implicit reasoning between user queries and object content to determine relevance. As a result, traditional retrieval methods, based on lexical matching and embedding similarity, struggle with such tasks. Existing solutions have tried to address this issue by utilizing LLMs to assess object relevance *online* for each query, which is both time-consuming and costly. To overcome this, we introduce EnrichIndex which instead utilizes the LLM *offline*, using its reasoning skills to enrich each object in the corpus and build new retrieval indices. EnrichIndex significantly improves retrieval performance for complex document and table retrieval tasks while greatly reducing online costs and latency. Additionally, we demonstrate how EnrichIndex can complement existing online LLM-based re-rankers to achieve even greater performance. Overall, we hope that our findings will inspire future research into more efficient offline document and table expansion and to further enhance retrieval performance beyond that of existing query rewriting and re-ranking approaches.

## Acknowledgments

We gratefully acknowledge the support of DARPA ASKEM Award HR00112220042, the ARPA-H Biomedical Data Fabric project, grants from Liberty Mutual and Jane Street, the Office of Naval Research (ONR N00014-23-1-2364), and the Croucher Scholarship.

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

# A  Example of enriching a document

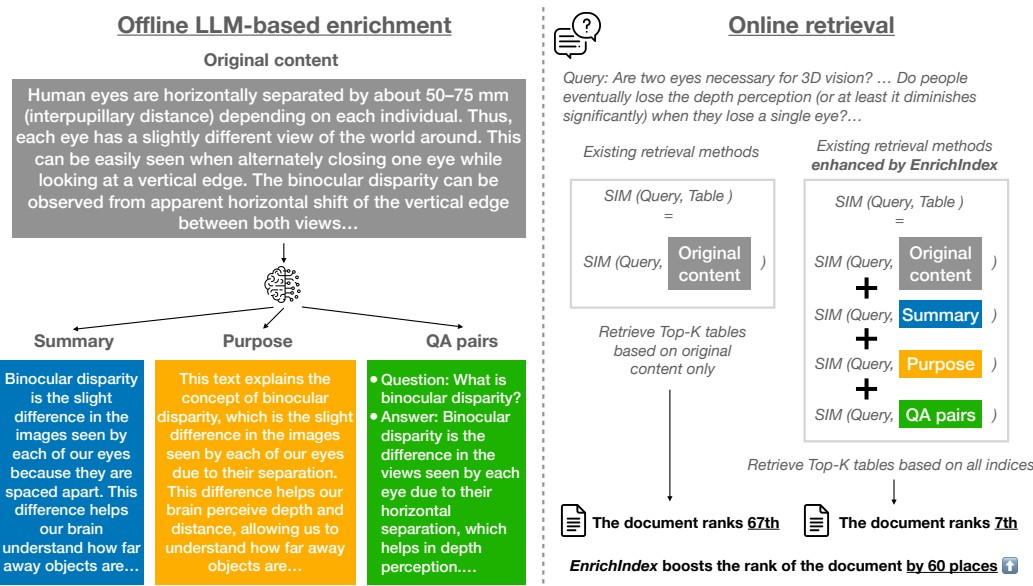

Figure 3: EnrichIndex leverages LLMs *offline* to enrich each object, creating multiple semantically-enhanced indices. During online retrieval, it computes object relevance by calculating a weighted sum of similarities between the user query across all enriched indices: the original document, its summary, purpose and QA pairs.

The user query focuses on "depth perception," a concept not explicitly stated in the original document, resulting in a low initial ranking of 67th by the Snowflake embedding model (Yu et al., 2024). However, the *summary*, *purpose*, and *QA pairs* introduced through enrichment include terms such as "far away," "depth and distance," and "depth perception," which better align with the query. As a result, the ranking improves significantly to 7th place. This demonstrates the effectiveness of the offline enrichment provided by EnrichIndex.

# B  Prompts

## B.1  Judgerank implmentation

JudgeRank was initially designed for the Bright dataset but not for table retrieval datasets, so we adapted it for table retrieval. In Bright, Judgerank utilizes GPT-4-0125-preview to generate expanded queries, but since the table datasets lack these expansions, we used the same prompt from the original Bright paper and employed GPT-4o-mini for query expansion. For question-specific object expansion, we applied similar prompts to tables as JudgeRank used for documents.

Tables 8 - 10 show the prompts used for query expansion, query-specific object expansion, and judging whether an object is relevant to the user query.

{User query}

Instructions:
1. Identify the essential problem.
2. Think step by step to reason and describe what information could be relevant and helpful to address the questions in detail.
3. Draft an answer with as many thoughts as you have.

Table 8: Zero-shot prompt for query expansion.

You will be presented with a/an {query name}, an analysis of the {query name}, and a/an {object name}.

Your task consists of the following steps:
1. Analyze the {object name}:
- Thoroughly examine each {unit} of the {object name}.
- List all {units} from the {object name} that {relevance} the {query name}.
- Briefly explain how each {unit} listed {relevance} the {query name}.

2. Assess overall relevance:
- If the {object name}, particularly the relevant {units} (if applicable), {relevance} the {query name}, briefly explain why.
- Otherwise, briefly explain why not.

Here is the {query name}:
{query}

Here is the analysis of the {query name}:
{query analysis}

Here is the {object name}:
{object}

Table 9: Zero-shot prompt for query-dependent object expansion.

You will be presented with a/an {query name}, an analysis of the {query name}, a/an {object name}, and an analysis of the {object name}.

Your task is to assess if the {object name} {relevance} the {query name} in one word:
- Yes: If the {object name} {relevance} the {query name}.
- No: Otherwise.

Important: Respond using only one of the following two words without quotation marks:
Yes or No.

Here is the {query name}:
{query}

Here is the analysis of the {query name}:
{query analysis}

Here is the {object name}:
{object}

Here is the analysis of the {object name}:
{object analysis}

Table 10: Zero-shot prompt for judgment.

Specific values for `query name`, `object name`, and `relevance` for each dataset used in the prompts above.

```
{
  "biology": {
    "query name": "Biology post",
    "object name": "document",
    "relevance": "substantially helps answer"
  },
  "earth_science": {
    "query name": "Earth Science post",
    "object name": "document",
    "relevance": "substantially helps answer"
  },
  "economics": {
    "query name": "Economics post",
    "object name": "document",
    "relevance": "substantially helps answer"
  },
  "psychology": {
    "query name": "Psychology post",
    "object name": "document",
    "relevance": "substantially helps answer"
  },
  "robotics": {
    "query name": "Robotics post",
    "object name": "document",
    "relevance": "substantially helps answer"
  },
  "stackoverflow": {
    "query name": "Stack Overflow post",
    "object name": "document",
    "relevance": "substantially helps answer"
  },
  "sustainable_living": {
```

```
    "query name": "Sustainable Living post",
    "object name": "document",
    "relevance": "substantially helps answer"
  },
  "leetcode": {
    "query name": "coding problem",
    "object name": "solved coding problem",
    "relevance": "uses the same algorithmic design"
  },
  "pony": {
    "query name": "Pony coding problem",
    "object name": "documentation",
    "relevance": "contains the required syntax for solving"
  },
  "aops": {
    "query name": "Math problem",
    "object name": "solved Math problem",
    "relevance": "uses the same problem-solving skill as"
  },
  "theoremqa_questions": {
    "query name": "Math problem",
    "object name": "solved math problem",
    "relevance": "uses the same theorem as"
  },
  "theoremqa_theorems": {
    "query name": "Math problem",
    "object name": "theorem",
    "relevance": "substantially helps answer"
  },
  "spider2": {
    "query name": "user query",
    "object name": "table",
    "relevance": "substantially helps answer"
  },
  "beaver": {
    "query name": "user query",
    "object name": "table",
    "relevance": "substantially helps answer"
  },
  "fiben": {
    "query name": "user query",
    "object name": "table",
    "relevance": "substantially helps answer"
  }
}
```

### B.2 Prompts for object enrichment

Tables 11 - 13 show the prompts used for object enrichment for each dataset.

*// Purpose*
Given the following technical text, describe the purpose of this text in layman's terms in one paragraph. If you do not think the text is semantically meaningful, output None.
*// Summary*
Given the following technical text, summarize this text in layman's terms in one paragraph. If you do not think the text is semantically meaningful, output None.
*// QA pairs*
Given the following technical text, generate at most 20 distinct question-answer pairs on this text. The questions should be general, and phrased in layman's terms, using vocabulary that can be distinct from the text, but still requires explicit or implicit knowledge from the text. Each question-answer pair should be formatted as a list where the first element is the question and the second element is the answer. The output should be a list of lists in JSON format. If you do not think the text is semantically meaningful, output None.

{document...}

Table 11: Zero-shot prompt for document enrichment in the Bright dataset.

*// Purpose*
Given the following text, describe the purpose of this text in layman's terms in one paragraph. If you do not think the text is semantically meaningful, output None.
*// Summary*
Given the following text, summarize this text in layman's terms in one paragraph. If you do not think the text is semantically meaningful, output None.
*// QA pairs*
Given the following text, generate at most 20 distinct question-answer pairs on this text. The questions should be general, and phrased in layman's terms, using vocabulary that can be distinct from the text, but still requires explicit or implicit knowledge from the text. Each question-answer pair should be formatted as a list where the first element is the question and the second element is the answer. The output should be a list of lists in JSON format. If you do not think the text is semantically meaningful, output None.

{document...}

Table 12: Zero-shot prompt for document enrichment in the NQ dataset.

*// Purpose*
Given the following table, describe the purpose of this table in layman's terms in one paragraph. If you do not think the text is semantically meaningful, output None.
*// Summary*
Given the following table, summarize this table in layman's terms in one paragraph. If you do not think the text is semantically meaningful, output None.
*// QA pairs*
Given the following table, generate at most 20 distinct question-answer pairs on this table that includes both simple questions and those requiring summarization or aggregation. The questions should be phrased in layman's terms, using explicit or implicit knowledge from the table. The question and answer should avoid using exact terms, such as shortened forms, from the table but can instead use naturally phrased language. Each question-answer pair should be formatted as a list where the first element is the question and the second element is the answer. The output should be a list of lists in JSON format. If you do not think the text is semantically meaningful, output None.

{table...}

Table 13: Zero-shot prompt for table enrichment for the table retrieval datasets.

## C Datasets

We conducted evaluations using the standard test sets for each benchmark, except for Spider2, where we used the publicly available dev set. This choice was made because Spider2 only provides gold tables for each user query in the dev set, which are essential for assessing retrieval performance.

Overall, the dataset consists of 1,384 questions and 1,145,164 documents for Bright, 258 questions and 5,088 tables for Spider2, 209 questions and 463 tables for Beaver, 300 questions and 152 tables for Fiben, and 3,452 questions and 500,000 documents for NQ.

## D Table serialization

Following Chen et al. (2025); Lei et al. (2024), tables were formatted as markdown, including database names, column names, and a sample of five randomly selected rows. An example is provided below.

```
Database name: META_KAGGLE
Table name: META_KAGGLE.META_KAGGLE.KERNELVERSIONCOMPETITIONSOURCES
Example table content:
| Id | KernelVersionId | SourceCompetitionId |
|-------:|------------------:|----------------------:|
| 6280 | 4511 | 3948 |
| 601723 | 555 | 3948 |
| 88411 | 841 | 3948 |
| 974621 | 610 | 3948 |
| 950711 | 773 | 3948 |
```

## E Hyperparameter tuning

We split each dataset into an 80/20 ratio, with 80% designated for testing and 20% for validation. The validation set was used to fine-tune the coefficients for our method and the hybrid approach.

## F   Detailed stage-one retrieval performance

| | \multicolumn{8}{c}{$k = 10$} | | | | | | | | \multicolumn{8}{c}{$k = 100$} | | | | | | | |
|---|---|---|---|---|---|---|---|---|---|---|---|---|---|---|---|---|
| | StackEx. | | Coding | | Thm. | | Avg. | | StackEx. | | Coding | | Thm. | | Avg. | |
| | R | N | R | N | R | N | R | N | R | N | R | N | R | N | R | N |
| *Original question* | | | | | | | | | | | | | | | | |
| BM25 | 23.4 | 18.9 | 17.1 | 10.4 | 5.5 | 3.8 | 17.3 | 13.2 | 48.1 | 25.8 | 31.7 | 15.0 | 14.1 | 6.0 | 35.7 | 18.4 |
| BM25$_E$ | 21.7 | 17.6 | 14.8 | 9.8 | 5.8 | 4.0 | 16.0 | 12.4 | 49.7 | 25.6 | 31.8 | 14.8 | 14.6 | 6.2 | 36.8 | 18.3 |
| UAE | 19.6 | 16.5 | 15.4 | 10.5 | 8.3 | 5.5 | 15.7 | 12.4 | 47.4 | 24.2 | 32.7 | 15.7 | 19.7 | 8.4 | 37.1 | 18.3 |
| UAE$_E$ | 22.9 | 19.1 | 18.5 | 14.1 | 8.8 | 6.2 | 18.2 | 14.6 | 52.1 | 27.3 | 35.3 | 18.9 | 23.7 | 9.8 | 41.2 | 20.9 |
| GTE | 18.7 | 15.0 | 13.9 | 11.9 | 9.2 | 6.1 | 15.2 | 11.9 | 47.8 | 23.1 | 37.6 | 18.9 | 24.3 | 10.0 | 39.4 | 18.7 |
| GTE$_E$ | 20.9 | 16.6 | 18.6 | 15.0 | 13.4 | 9.3 | 18.4 | 14.3 | 52.3 | 25.4 | 43.8 | 22.9 | 27.4 | 12.7 | 43.9 | 21.5 |
| Snow. | 22.3 | 18.9 | 15.6 | 10.4 | 10.4 | 6.8 | 17.8 | 14.0 | 46.7 | 25.6 | 34.7 | 15.9 | 24.2 | 10.2 | 38.3 | 19.6 |
| Snow.$_E$ | 25.7 | 21.6 | 20.1 | 16.9 | 12.1 | 7.7 | 20.9 | 16.9 | 56.0 | 30.2 | 46.4 | 24.4 | 27.8 | 11.7 | 46.4 | 24.0 |
| BM25+UAE | 23.9 | 20.1 | 18.0 | 12.5 | 8.9 | 5.7 | 18.7 | 14.7 | 53.6 | 28.5 | 37.8 | 18.6 | 22.8 | 9.3 | 42.2 | 21.4 |
| (BM25+UAE)$_E$ | 25.4 | 21.4 | 17.4 | 13.4 | 9.7 | 6.4 | 19.6 | 15.8 | 58.4 | 30.6 | 38.8 | 19.6 | 24.5 | 9.9 | 45.4 | 22.8 |
| BM25+GTE | 25.6 | 21.0 | 18.7 | 15.5 | 11.6 | 7.3 | 20.5 | 16.2 | 54.7 | 29.2 | 44.6 | 23.3 | 26.7 | 11.1 | 45.1 | 23.1 |
| (BM25+GTE)$_E$ | 26.8 | 22.1 | 20.7 | 16.8 | 13.2 | 9.1 | 21.9 | 17.5 | 57.4 | 30.5 | 47.8 | 24.8 | 29.8 | 13.3 | 48.0 | 24.7 |
| BM25+Snow. | 27.2 | 22.2 | 17.9 | 12.1 | 10.4 | 6.9 | 20.8 | 16.1 | 54.7 | 30.1 | 40.6 | 18.8 | 24.5 | 10.5 | 43.8 | 22.6 |
| (BM25+Snow.)$_E$ | 27.2 | 21.9 | 20.8 | 17.3 | 12.0 | 7.9 | 21.9 | 17.2 | 59.6 | 31.3 | 47.4 | 24.8 | 28.0 | 11.9 | 48.7 | 24.8 |
| Average | 23.0 | 18.9 | 16.7 | 11.9 | 9.2 | 6.0 | 18.0 | 14.1 | 50.4 | 26.7 | 37.1 | 18.0 | 22.3 | 9.4 | 40.2 | 20.3 |
| Average$_E$ | 24.4 | 20.0 | 18.7 | 14.8 | 10.7 | 7.2 | 19.6 | 15.5 | 55.1 | 28.7 | 41.6 | 21.5 | 25.1 | 10.8 | 44.3 | 22.4 |
| *GPT-4 generated expanded query* | | | | | | | | | | | | | | | | |
| BM25 | 37.7 | 32.8 | 12.2 | 9.5 | 13.7 | 9.8 | 26.4 | 22.2 | 64.4 | 40.2 | 26.4 | 13.8 | 27.8 | 13.6 | 47.3 | 28.0 |
| BM25$_E$ | 38.2 | 33.5 | 12.9 | 9.4 | 19.8 | 13.5 | 28.5 | 23.6 | 67.8 | 41.7 | 25.3 | 12.9 | 34.7 | 17.5 | 50.8 | 29.7 |
| UAE | 28.2 | 25.0 | 15.6 | 13.5 | 16.6 | 11.7 | 22.7 | 19.2 | 55.4 | 32.5 | 38.0 | 20.4 | 31.9 | 15.7 | 45.7 | 25.6 |
| UAE$_E$ | 29.4 | 25.6 | 18.1 | 17.1 | 18.5 | 13.6 | 24.3 | 20.7 | 60.8 | 34.3 | 42.9 | 24.0 | 34.3 | 17.6 | 50.2 | 27.8 |
| GTE | 24.5 | 20.1 | 13.1 | 11.2 | 15.0 | 10.6 | 19.8 | 15.9 | 52.9 | 28.0 | 34.5 | 17.8 | 28.2 | 13.9 | 42.7 | 22.2 |
| GTE$_E$ | 25.5 | 21.7 | 16.4 | 15.6 | 18.4 | 13.1 | 21.9 | 18.2 | 56.2 | 30.2 | 40.5 | 22.5 | 32.0 | 16.6 | 46.6 | 25.0 |
| Snow. | 32.9 | 27.9 | 16.1 | 10.3 | 16.6 | 12.1 | 25.3 | 20.3 | 60.1 | 35.3 | 34.4 | 15.6 | 32.1 | 15.8 | 47.7 | 26.3 |
| Snow.$_E$ | 32.7 | 28.3 | 18.7 | 16.4 | 18.4 | 13.0 | 26.2 | 21.9 | 64.2 | 36.9 | 48.8 | 25.2 | 36.6 | 17.6 | 53.8 | 29.4 |
| BM25+UAE | 36.1 | 32.1 | 17.3 | 14.1 | 17.6 | 12.6 | 27.5 | 23.4 | 66.1 | 40.5 | 40.2 | 21.1 | 32.6 | 16.3 | 52.1 | 30.3 |
| (BM25+UAE)$_E$ | 37.0 | 32.5 | 17.4 | 16.7 | 20.3 | 14.4 | 28.8 | 24.6 | 66.4 | 40.8 | 44.5 | 24.2 | 37.4 | 18.6 | 54.4 | 31.6 |
| BM25+GTE | 37.2 | 33.0 | 18.5 | 15.6 | 16.6 | 11.8 | 28.1 | 23.9 | 66.3 | 41.1 | 40.7 | 22.9 | 29.0 | 15.1 | 51.3 | 30.5 |
| (BM25+GTE)$_E$ | 39.5 | 34.7 | 19.0 | 17.2 | 19.8 | 14.5 | 30.3 | 25.9 | 70.8 | 43.2 | 45.7 | 25.1 | 34.9 | 18.4 | 56.3 | 33.1 |
| BM25+Snow. | 37.9 | 33.0 | 15.8 | 10.6 | 17.7 | 13.0 | 28.3 | 23.4 | 66.9 | 41.1 | 32.7 | 16.0 | 33.9 | 17.1 | 51.5 | 29.8 |
| (BM25+Snow.)$_E$ | 39.7 | 34.7 | 20.2 | 17.0 | 20.7 | 14.4 | 30.9 | 25.8 | 67.9 | 42.5 | 48.9 | 25.5 | 36.4 | 18.5 | 55.7 | 32.8 |
| Average | 33.5 | 29.1 | 15.5 | 12.1 | 16.3 | 11.7 | 25.4 | 21.2 | 61.7 | 37.0 | 35.3 | 18.2 | 30.8 | 15.3 | 48.3 | 27.6 |
| Average$_E$ | 34.6 | 30.1 | 17.5 | 15.6 | 19.4 | 13.8 | 27.3 | 23.0 | 64.9 | 38.5 | 42.4 | 22.8 | 35.2 | 17.8 | 52.5 | 29.9 |

Table 14: Recall (R) and NDCG (N) @ $k$ of stage-one retrievers on the Bright dataset.

| | k = 10 | | | | | | | | k = 20 | | | | | | | |
| | Spider2 | | Beaver | | Fiben | | Avg. | | Spider2 | | Beaver | | Fiben | | Avg. | |
| | R | N | R | N | R | N | R | N | R | N | R | N | R | N | R | N |
| *Original question* | | | | | | | | | | | | | | | | |
| BM25 | 43.3 | 32.0 | 51.7 | 43.9 | 30.6 | 29.0 | 40.6 | 34.1 | 53.0 | 35.1 | 67.7 | 50.0 | 33.8 | 30.2 | 49.5 | 37.2 |
| BM25$_E$ | 60.9 | 46.5 | 59.2 | 52.4 | 47.4 | 40.9 | 55.2 | 45.9 | 69.8 | 49.6 | 73.9 | 58.4 | 62.9 | 46.6 | 68.2 | 50.8 |
| UAE | 55.4 | 44.2 | 48.6 | 43.8 | 48.2 | 42.6 | 50.7 | 43.5 | 70.2 | 48.9 | 67.5 | 51.2 | 61.4 | 47.8 | 66.0 | 49.1 |
| UAE$_E$ | 60.2 | 47.0 | 54.8 | 49.3 | 55.3 | 52.0 | 56.8 | 49.6 | 74.2 | 51.7 | 72.4 | 56.3 | 63.3 | 55.3 | 69.5 | 54.3 |
| GTE | 53.8 | 41.9 | 54.0 | 45.9 | 54.4 | 56.9 | 54.1 | 48.8 | 62.4 | 44.7 | 69.6 | 52.2 | 63.1 | 60.4 | 64.7 | 52.9 |
| GTE$_E$ | 61.5 | 47.0 | 58.0 | 51.7 | 60.7 | 62.7 | 60.2 | 54.4 | 72.5 | 50.7 | 74.0 | 58.1 | 67.2 | 65.3 | 70.8 | 58.4 |
| Snowflake | 45.1 | 34.1 | 49.2 | 42.3 | 51.8 | 47.5 | 48.8 | 41.6 | 53.1 | 36.5 | 66.3 | 49.3 | 58.7 | 50.4 | 58.9 | 45.4 |
| Snowflake$_E$ | 62.0 | 48.1 | 58.0 | 52.4 | 53.9 | 53.8 | 57.7 | 51.5 | 71.3 | 51.0 | 74.1 | 59.0 | 59.4 | 56.0 | 67.4 | 55.1 |
| BM25+UAE | 65.2 | 50.7 | 55.3 | 48.0 | 44.8 | 37.8 | 54.5 | 45.0 | 74.2 | 53.9 | 76.2 | 56.2 | 61.5 | 44.3 | 69.8 | 50.8 |
| (BM25+UAE)$_E$ | 67.7 | 53.0 | 62.9 | 54.2 | 52.1 | 43.2 | 60.3 | 49.5 | 77.7 | 56.4 | 79.2 | 60.8 | 63.1 | 47.5 | 72.4 | 54.1 |
| BM25+GTE | 60.2 | 48.1 | 57.1 | 49.1 | 51.8 | 43.8 | 56.1 | 46.7 | 69.4 | 51.1 | 77.2 | 56.9 | 62.5 | 48.1 | 68.8 | 51.5 |
| (BM25+GTE)$_E$ | 68.1 | 52.5 | 64.7 | 56.5 | 58.6 | 48.3 | 63.5 | 51.9 | 76.1 | 55.2 | 80.7 | 62.8 | 66.9 | 51.6 | 73.8 | 55.9 |
| BM25+Snow. | 55.1 | 42.8 | 55.5 | 47.0 | 50.3 | 42.0 | 53.4 | 43.6 | 64.3 | 45.8 | 75.2 | 54.7 | 59.3 | 45.5 | 65.3 | 48.1 |
| (BM25+Snow.)$_E$ | 68.7 | 53.0 | 63.9 | 55.5 | 50.6 | 44.9 | 60.3 | 50.5 | 77.8 | 55.9 | 80.4 | 62.1 | 59.2 | 48.3 | 71.2 | 54.6 |
| Average | 54.0 | 42.0 | 53.1 | 45.7 | 47.4 | 42.8 | 51.2 | 43.3 | 63.8 | 45.1 | 71.4 | 52.9 | 57.2 | 46.7 | 63.3 | 47.9 |
| Average$_E$ | 64.1 | 49.6 | 60.2 | 53.1 | 54.1 | 49.4 | 59.1 | 50.5 | 74.2 | 52.9 | 76.4 | 59.6 | 63.1 | 52.9 | 70.5 | 54.7 |
| *GPT-4o-mini generated expanded query* | | | | | | | | | | | | | | | | |
| BM25 | 39.1 | 28.6 | 47.0 | 37.5 | 46.2 | 34.5 | 44.0 | 33.3 | 49.3 | 31.8 | 65.3 | 44.8 | 76.9 | 46.0 | 64.5 | 40.9 |
| BM25$_E$ | 60.4 | 46.5 | 58.4 | 51.9 | 61.2 | 46.2 | 60.2 | 47.9 | 71.8 | 50.3 | 75.3 | 58.7 | 79.2 | 53.2 | 75.7 | 53.7 |
| UAE | 56.8 | 42.4 | 47.7 | 41.7 | 49.3 | 45.9 | 51.4 | 43.6 | 68.7 | 46.0 | 65.2 | 48.5 | 67.2 | 52.6 | 67.2 | 49.3 |
| UAE$_E$ | 61.8 | 47.8 | 53.8 | 48.0 | 55.1 | 52.7 | 57.0 | 49.8 | 73.1 | 51.5 | 68.9 | 54.0 | 63.5 | 56.0 | 68.2 | 54.0 |
| GTE | 52.1 | 43.2 | 49.7 | 41.2 | 57.4 | 57.4 | 53.5 | 48.2 | 64.3 | 47.1 | 63.9 | 46.8 | 66.8 | 61.2 | 65.2 | 52.5 |
| GTE$_E$ | 56.5 | 44.7 | 55.9 | 47.9 | 62.2 | 62.8 | 58.6 | 52.6 | 66.4 | 48.0 | 70.7 | 53.8 | 70.4 | 66.0 | 69.1 | 56.6 |
| Snowflake | 49.7 | 37.2 | 51.1 | 44.3 | 53.3 | 52.8 | 51.5 | 45.3 | 57.4 | 39.6 | 65.9 | 50.3 | 60.3 | 55.7 | 60.9 | 48.8 |
| Snowflake$_E$ | 62.9 | 47.6 | 52.6 | 45.7 | 55.4 | 56.0 | 57.1 | 50.4 | 73.6 | 51.1 | 66.9 | 51.5 | 61.6 | 58.5 | 67.1 | 54.1 |
| BM25+UAE | 64.0 | 47.9 | 54.8 | 47.4 | 50.8 | 42.3 | 56.3 | 45.6 | 73.5 | 50.8 | 71.7 | 54.1 | 73.8 | 50.9 | 73.1 | 51.8 |
| (BM25+UAE)$_E$ | 67.8 | 52.8 | 60.8 | 55.3 | 62.7 | 47.6 | 63.9 | 51.5 | 79.5 | 56.8 | 75.9 | 61.5 | 78.6 | 53.6 | 78.2 | 56.8 |
| BM25+GTE | 58.8 | 47.1 | 56.2 | 46.1 | 59.7 | 44.2 | 58.4 | 45.7 | 66.8 | 49.8 | 74.2 | 53.2 | 81.0 | 52.2 | 74.4 | 51.6 |
| (BM25+GTE)$_E$ | 66.4 | 51.7 | 62.1 | 55.8 | 65.7 | 51.5 | 65.0 | 52.7 | 76.5 | 55.0 | 76.2 | 61.5 | 80.3 | 57.1 | 77.9 | 57.6 |
| BM25+Snow. | 55.7 | 43.7 | 56.6 | 49.0 | 57.4 | 42.3 | 56.6 | 44.6 | 64.4 | 46.6 | 74.0 | 56.0 | 74.3 | 48.8 | 70.9 | 50.0 |
| (BM25+Snow.)$_E$ | 66.1 | 52.8 | 61.5 | 56.2 | 61.2 | 46.2 | 63.0 | 51.1 | 75.2 | 55.8 | 75.0 | 61.8 | 79.2 | 53.2 | 76.7 | 56.4 |
| Average | 53.8 | 41.5 | 51.9 | 43.9 | 53.4 | 45.6 | 53.1 | 43.7 | 63.5 | 44.5 | 68.6 | 50.5 | 71.5 | 52.5 | 68.0 | 49.3 |
| Average$_E$ | 63.1 | 49.1 | 57.9 | 51.5 | 60.5 | 51.9 | 60.7 | 50.9 | 73.7 | 52.6 | 72.7 | 57.5 | 73.3 | 56.8 | 73.3 | 55.6 |

Table 15: Recall and NDCG @ $k$ of various methods on the table retrieval datasets.

| | k = 10 | | k = 100 | |
| | Recall | NDCG | Recall | NDCG |
| --- | --- | --- | --- | --- |
| BM25 | 69.4 | 51.8 | 87.4 | 55.9 |
| BM25$_E$ | 74.9 | 57.6 | 90.6 | 61.2 |
| UAE | 91.5 | 77.0 | 98.1 | 78.6 |
| UAE$_E$ | 91.5 | 77.7 | 98.3 | 79.4 |
| GTE | 88.0 | 73.8 | 97.0 | 75.9 |
| GTE$_E$ | 90.5 | 76.4 | 98.0 | 78.2 |
| Snow. | 94.6 | 82.7 | 98.7 | 83.6 |
| Snow.$_E$ | 95.1 | 83.2 | 99.2 | 84.2 |
| BM25+UAE | 91.5 | 77.0 | 98.1 | 78.6 |
| (BM25+UAE)$_E$ | 91.5 | 77.7 | 98.3 | 79.4 |
| BM25+GTE | 89.0 | 74.6 | 97.8 | 76.6 |
| (BM25+GTE)$_E$ | 90.7 | 75.8 | 98.1 | 77.5 |
| BM25+Snow. | 94.6 | 82.7 | 98.7 | 83.6 |
| (BM25+Snow.)$_E$ | 95.1 | 83.2 | 99.2 | 84.2 |
| Average | 88.4 | 74.2 | 96.5 | 76.1 |
| Average$_E$ | 89.9 | 76.0 | 97.4 | 77.7 |

Table 16: Recall and NDCG @ $k$ of stage-one retrievers on the NQ dataset.

# G    LLM usage of different methods

|  | StackExchange | Coding | Theorems | Average | Spider2 | Beaver | Fiben | Average |
|---|---|---|---|---|---|---|---|---|
| **EnrichIndex** | | | | | | | | |
| #Input tokens | 343.4 | 436.6 | 188.7 | 317.8 | 139.0 | 138.0 | 82.7 | 116.7 |
| #Output tokens | 850.9 | 867.3 | 913.6 | 871.2 | 929.0 | 998.1 | 827.0 | 907.9 |
| #Total tokens | 1194.3 | 1304.0 | 1102.2 | 1189.0 | 1068.0 | 1136.1 | 909.7 | 1024.6 |
| **Judgerank** | | | | | | | | |
| #Input tokens | 368453.9 | 380929.8 | 359144.0 | 368172.3 | 558255.9 | 102817.2 | 44932.8 | 233205.7 |
| #Output tokens | 47691.9 | 46343.1 | 53994.3 | 49184.6 | 8575.6 | 10005.0 | 6389.1 | 8109.0 |
| #Total tokens | 416145.8 | 427272.9 | 413138.3 | 417356.9 | 566831.6 | 112822.2 | 51321.9 | 241314.6 |

Table 17:  Absolute number of input, output, and total tokens used by LLMs for EnrichIndex-enhanced stage-one retriever and Judgerank. Lower token counts signify reduced latency and cost.

