# OpenReview forum: "EnrichIndex: Using LLMs to Enrich Retrieval Indices Offline"
_colmweb.org/COLM/2025/Conference — COLM 2025_

### Official Review · Reviewer_Cq8Z · 2025-05-09

**Rating:** 7
**Confidence:** 4
**Ethics Flag:** 1

**Summary:**

Enrichindex proposes to augment documents with a summary, purpose, and QA pairs to improve the performance of first and second stage retrieval. The document augmentation is done with GPT-4o-mini. Each artifact was indexed separately so that a dataset independent weighting could be used to combine the document scores from the original and each independent index. The approach is evaluated on the Bright dataset and on a number of table retrieval datasets, where tables were combined across datasets to create a single table corpus.

While the approach is straightforward and improves performance on dense retrievers on Bright and all first stage retrievers for tables, the usefulness for re-rankers is less convincing. This is likely because the reranker (based on Llama-3.1-8B-Instruct) is far less capable than the model used for document enrichment.

A second weakness is that the paper includes no related work section. Thus it does not compare to other document enrichment approaches such as doc2query. The paper needs to situate this work within the body of literature on document expansion. It needs to identify its contributions in this context.

Another odd choice is report nDCG@100. This score is rarely reported because nDCG tends to give too much credit to low ranked items. Therefore, one should not report nDCG at ranks below which you would not expect a user to look. This is why rank 10 is a very common cutoff. Reporting lower-levels of recall especially for first stage retrievers is acceptable because once could apply a reranker to a lower depth than a user would look in an effort to raise relevant documents into the top 10 such as recall at 100 or even recall at 1000 if you would re-rank to that depth.


While demonstrating that the added information is about to move documents closer to the query using Wasserstein distance in Section 4.1 is somewhat interesting, this fails to capture the interaction between relevant and non-relevant information that is key to ranking relevant information above non-relevant information. It is not sufficient to move relevant information closer to the query if non-relevant information also moves closer to the query. Exploring the dependence on learning the weights of these artifacts in EnrichIndex would be a better use of the space. If good performance is very sensitive to the weighting the approach may be less useful because many search engines are expected to retrieve a wide variety of data.

Finally reporting on a subset of Bright makes it appear that the dataset was cherry picked for the best subset of the technique. The remaining Bright numbers should at least appear in the appendix for the interested reader.

**Questions To Authors:**

Q1 How well does the approach work on the remaining Bright tasks?
Q2 Can you provide a generic setting of a1,a2, a3, and a4 to use as a comparison to the tuned version to help inform the reader about the robustness of the approach?

**Reasons To Accept:**

+ The paper appears to explore some innovative approaches for document expansion for retrieval
+ The approach is particularly effective for tables which are generally a challenging domain because of the lack of context for the information

**Reasons To Reject:**

- Lack of context of this work relative to prior work on document expansion
- Unfair comparison with rerankers
- Unclear how dependent the approach is on setting the different weights for the final ranking function

---

> ### Author Response · Authors · 2025-06-02
> **Author Response (Part 4/4)**
>
> > Q1 How well does the approach work on the remaining Bright tasks?
>
> As detailed above, the performance is reported for **all** tasks in Bright.
>
> > Q2 Can you provide a generic setting of a1,a2, a3, and a4 to use as a comparison to the tuned version to help inform the reader about the robustness of the approach?
>
> The weight tuning analysis above demonstrates that our approach maintains strong performance even when using limited data or a single set of weights across multiple datasets, underscoring its robustness and generalizability. Moreover, the tuning process is highly parallelizable, requires minimal effort, and enables easy adaptation of our general retrieval framework to various workloads.

---

> > ### Comment · Reviewer_Cq8Z · 2025-06-07
> > **Single Weight Set**
> >
> > You indicate that you have a single weight set and it performs well; however, it is unclear to me where those results are. Could  you point me to them by Table if they are in the original paper or here in this response. Also, what are those values. They need to appear somewhere in the paper.

---

> > > ### Author Response · Authors · 2025-06-07
> > >
> > > The results are reported in the lower half of *Author Response (Part 3/4 continued)*. We copy it here.
> > >
> > > In addition, we explored whether a single set of coefficients could be tuned and applied across multiple datasets. To test this, we combined the validation sets from the three table retrieval datasets—Spider2, Beaver, and Fiben—into one unified validation set for the weight tuning process. The results, summarized in the table below, show that EnrichIndex still delivers substantial performance improvements, demonstrating the generalizability of our approach across diverse workloads simultaneously.
> > >
> > > | Original query                   | Spider2 (R) | Spider2 (N) | Beaver (R) | Beaver (N) | Fiben (R) | Fiben (N) | Average (R) | Average (N) |
> > > |--------------------------|-------------|-------------|-------------|-------------|------------|------------|---------------|---------------|
> > > | Average                  | 54.0        | 41.9        | 53.5        | 45.7        | 47.3       | 42.7       | 51.2          | 43.2          |
> > > | Average with EnrichIndex | **64.2**        | **50.4**        | **59.6**        | **52.3**        | **51.7**       | **47.8**       | **58.0**          | **49.9**          |
> > >
> > >
> > > | GPT-4 generated expanded query                   | Spider2 (R) | Spider2 (N) | Beaver (R) | Beaver (N) | Fiben (R) | Fiben (N) | Average (R) | Average (N) |
> > > |--------------------------|-------------|-------------|-------------|-------------|------------|------------|---------------|---------------|
> > > | Average                  | 53.6        | 41.3        | 52.1        | 44.0        | 53.6       | 46.5       | 53.2          | 44.1          |
> > > | Average with EnrichIndex | **63.7**        | **49.4**       | **57.5**        | **51.5**        | **59.9**       | **54.2**       | **60.5**          | **51.9**          |

---

> > > > ### Comment · Reviewer_Cq8Z · 2025-06-08
> > > >
> > > > What are the actual weights and do they also work for bright?

---

> > > > > ### Author Response · Authors · 2025-06-09
> > > > >
> > > > > Following the reviewer’s comments, we further investigate the potential of a *single* general-purpose set of EnrichIndex weights across different tasks. Namely, we combine the validation sets for *all* datasets, Bright, Spider2, Beaver, and Fiben, into one unified validation set for the weight tuning process. The results, summarized in the table below, show that EnrichIndex still delivers substantial performance improvements, demonstrating the generalizability of our approach across diverse workloads simultaneously. We will include these additional results in the next version of our paper.
> > > > >
> > > > > The weights are (0.4, 0.2, 0.2, 0.2) in the order of (original content, purpose, summary, QA pairs).
> > > > >
> > > > > R stands for recall @ 10, N stands for NDCG @ 10
> > > > >
> > > > > | Original query                   | StackExchange (R) | StackExchange (N) | Coding (R) | Coding (N) | Theorems (R) | Theorems (N) | Average (R) | Average (N) |
> > > > > |--------------------------|-------------------|--------------------|------------|-------------|----------------|----------------|--------------|--------------|
> > > > > | Average                  | 22.6              | 18.7               | 16.5       | 11.8        | 9.3            | 6.2            | 17.8         | 13.9         |
> > > > > | Average with EnrichIndex | **24.1**              | **20.0**               | **17.8**       | **13.5**        | **10.8**           | **7.3**            | **19.3**         | **15.3**        |
> > > > >
> > > > >
> > > > > | GPT-4 generated expanded query                   | StackExchange (R) | StackExchange (N) | Coding (R) | Coding (N) | Theorems (R) | Theorems (N) | Average (R) | Average (N) |
> > > > > |--------------------------|-------------------|--------------------|------------|-------------|----------------|----------------|--------------|--------------|
> > > > > | Average                  | 33.5              | 29.3               | 15.6       | 12.2        | 16.8           | 11.9           | 25.6         | 21.4         |
> > > > > | Average with EnrichIndex | **34.7**              | **30.3**               | **17.2**       | **14.3**        | **19.5**           | **14.0**          | **27.3**         | **22.9**         |
> > > > >
> > > > >
> > > > > | Original query                   | Spider2 (R) | Spider2 (N) | Beaver (R) | Beaver (N) | Fiben (R) | Fiben (N) | Average (R) | Average (N) |
> > > > > |--------------------------|-------------|-------------|-------------|-------------|------------|------------|---------------|---------------|
> > > > > | Average                  | 53.8        | 41.5        | 53.5        | 45.8        | 47.1       | 42.6       | 51.1          | 43.1         |
> > > > > | Average with EnrichIndex | **63.7**        | **50.1**        | **58.2**        | **51.0**        | **52.2**       | **47.6**       | **57.7**          | **49.3**          |
> > > > >
> > > > > | GPT-4 generated expanded query                   | Spider2 (R) | Spider2 (N) | Beaver (R) | Beaver (N) | Fiben (R) | Fiben (N) | Average (R) | Average (N) |
> > > > > |--------------------------|-------------|-------------|-------------|-------------|------------|------------|---------------|---------------|
> > > > > | Average                  | 52.2        | 40.0        | 51.9        | 43.7        | 53.2       | 45.1       | 52.5          | 43.0          |
> > > > > | Average with EnrichIndex | **62.8**        | **48.9**        | **56.7**        | **50.4**        | **57.6**       | **50.0**       | **59.1**          | **49.8**          |

---

> > > > > > ### Comment · Reviewer_Cq8Z · 2025-06-09
> > > > > >
> > > > > > Thank you very much for these additional experiments. I look forward to seeing the next version of your paper and will raise my score to 7.

---

> ### Author Response · Authors · 2025-06-02
> **Author Response (Part 3/4 continued)**
>
> R stands for recall @ 10, N stands for NDCG @ 10
>
> | 10% validation data (original query)                   | StackExchange (R) | StackExchange (N) | Coding (R) | Coding (N) | Theorems (R) | Theorems (N) | Average (R) | Average (N) |
> |--------------------------|-------------------|--------------------|------------|-------------|----------------|----------------|--------------|--------------|
> | Average                  | 22.6              | 18.7               | 16.9       | 12.0        | 9.3            | 6.1            | 17.8         | 14.0         |
> | Average with EnrichIndex | **23.5**              | **19.3**               | **18.5**       | **14.9**        | **10.8**           | **7.2**            | **19.1**         | **15.2**        |
>
>
> | 10% validation data (GPT-4 generated expanded query)                   | StackExchange (R) | StackExchange (N) | Coding (R) | Coding (N) | Theorems (R) | Theorems (N) | Average (R) | Average (N) |
> |--------------------------|-------------------|--------------------|------------|-------------|----------------|----------------|--------------|--------------|
> | Average                  | 33.0              | 29.0               | 15.1       | 11.9        | 16.3           | 11.7           | 25.1         | 21.1         |
> | Average with EnrichIndex | **33.2**              | 28.9               | **16.9**       | **15.1**        | **19.5**           | **13.6**          | **26.5**         | **22.1**         |
>
>
> | 10% validation data (original query)                   | Spider2 (R) | Spider2 (N) | Beaver (R) | Beaver (N) | Fiben (R) | Fiben (N) | Average (R) | Average (N) |
> |--------------------------|-------------|-------------|-------------|-------------|------------|------------|---------------|---------------|
> | Average                  | 53.9        | 41.5        | 53.1        | 45.7        | 47.3       | 42.9       | 51.1          | 43.2          |
> | Average with EnrichIndex | **64.3**        | **49.2**        | **59.3**        | **52.7**        | **52.2**       | **47.8**       | **58.2**          | **49.6**          |
>
> | 10% validation data (GPT-4 generated expanded query)                   | Spider2 (R) | Spider2 (N) | Beaver (R) | Beaver (N) | Fiben (R) | Fiben (N) | Average (R) | Average (N) |
> |--------------------------|-------------|-------------|-------------|-------------|------------|------------|---------------|---------------|
> | Average                  | 53.6        | 41.5        | 52.0        | 43.7        | 52.9       | 46.1       | 52.9          | 43.9          |
> | Average with EnrichIndex | **62.9**        | **49.6**        | **56.8**        | **50.9**        | **58.9**       | **50.3**       | **59.7**          | **50.2**          |
>
>
> In addition, we explored whether a single set of coefficients could be tuned and applied across multiple datasets. To test this, we combined the validation sets from the three table retrieval datasets—Spider2, Beaver, and Fiben—into one unified validation set for the weight tuning process. The results, summarized in the table below, show that EnrichIndex still delivers substantial performance improvements, demonstrating the generalizability of our approach across diverse workloads simultaneously.
>
> | Original query                   | Spider2 (R) | Spider2 (N) | Beaver (R) | Beaver (N) | Fiben (R) | Fiben (N) | Average (R) | Average (N) |
> |--------------------------|-------------|-------------|-------------|-------------|------------|------------|---------------|---------------|
> | Average                  | 54.0        | 41.9        | 53.5        | 45.7        | 47.3       | 42.7       | 51.2          | 43.2          |
> | Average with EnrichIndex | **64.2**        | **50.4**        | **59.6**        | **52.3**        | **51.7**       | **47.8**       | **58.0**          | **49.9**          |
>
>
> | GPT-4 generated expanded query                   | Spider2 (R) | Spider2 (N) | Beaver (R) | Beaver (N) | Fiben (R) | Fiben (N) | Average (R) | Average (N) |
> |--------------------------|-------------|-------------|-------------|-------------|------------|------------|---------------|---------------|
> | Average                  | 53.6        | 41.3        | 52.1        | 44.0        | 53.6       | 46.5       | 53.2          | 44.1          |
> | Average with EnrichIndex | **63.7**        | **49.4**       | **57.5**        | **51.5**        | **59.9**       | **54.2**       | **60.5**          | **51.9**          |

---

> ### Author Response · Authors · 2025-06-02
> **Author Response (Part 3/4)**
>
> > Unclear how dependent the approach is on setting the different weights for the final ranking function
>
> Regarding the dependency of EnrichIndex on setting the weights: we demonstrate the robustness and generalizability of our approach, and add an analysis on the weight tuning process. Specifically, we evaluated whether a smaller portion of validation data could still yield strong performance. While the default setup uses 20% of the data for validation, we also experimented with using 15% and 10%, and reported the results for stage-one retrievers below. Overall, we observed that even with reduced validation data, EnrichIndex continues to deliver significant performance gains, highlighting the robustness of our tuning methodology. We will be sure to include this analysis in the final version of the paper.
>
> R stands for recall @ 10, N stands for NDCG @ 10
>
> | 15% validation data (original query)                   | StackExchange (R) | StackExchange (N) | Coding (R) | Coding (N) | Theorems (R) | Theorems (N) | Average (R) | Average (N) |
> |--------------------------|-------------------|--------------------|------------|-------------|----------------|----------------|--------------|--------------|
> | Average                  | 22.6              | 18.7               | 16.7       | 11.8        | 9.2            | 6.0            | 17.8         | 14.0         |
> | Average with EnrichIndex | **23.9**              | **19.4**               | **18.7**      | **14.6**       | **10.6**           | **7.2**            | **19.3**         | **15.2**         |
>
>
> | 15% validation data (GPT-4 generated expanded query)                   | StackExchange (R) | StackExchange (N) | Coding (R) | Coding (N) | Theorems (R) | Theorems (N) | Average (R) | Average (N) |
> |--------------------------|-------------------|--------------------|------------|-------------|----------------|----------------|--------------|--------------|
> | Average                  | 33.5              | 29.3               | 15.4       | 12.0        | 16.3           | 11.7           | 25.4         | 21.2         |
> | Average with EnrichIndex | **33.7**              | **29.4**               | **17.4**       | **15.5**        | **19.6**           | **13.8**           | **26.8**         | **22.5**        |
>
>
> | 15% validation data (original query)                   | Spider2 (R) | Spider2 (N) | Beaver (R) | Beaver (N) | Fiben (R) | Fiben (N) | Average (R) | Average (N) |
> | ------------------------ | ----------- | ----------- | ---------- | ---------- | --------- | --------- | ----------- | ----------- |
> | Average                  | 54.0        | 41.9        | 53.1       | 45.7       | 47.3      | 42.9      | 51.1        | 43.3        |
> | Average with EnrichIndex | **64.5**        | **49.9**        | **58.6**       | **52.4**       | **54.1**      | **49.3**      | **58.8**        | **50.3**        |
>
>
> | 15% validation data (GPT-4 generated expanded query)                   | Spider2 (R) | Spider2 (N) | Beaver (R) | Beaver (N) | Fiben (R) | Fiben (N) | Average (R) | Average (N) |
> | ------------------------ | ----------- | ----------- | ---------- | ---------- | --------- | --------- | ----------- | ----------- |
> | Average                  | 52.8        | 40.9        | 52.0       | 43.8       | 53.6      | 46.0      | 52.9        | 43.7        |
> | Average with EnrichIndex | **63.1**        | **49.4**        | **58.5**       | **52.1**       | **60.0**      | **51.6**      | **60.6**       | **51.0**        |

---

> > ### Comment · Reviewer_Cq8Z · 2025-06-07
> > **Average**
> >
> > In this table, could you clarify what the row Average refers to?

---

> > > ### Author Response · Authors · 2025-06-07
> > >
> > > *Average* refers to the average performance of all stage-one retrievers. The stage-one retrievers include (1) sparse retrievers (BM25), (2) dense retrievers (UAE, GTE, and Snowflake embedding models), and (3) a hybrid of sparse and dense retrievers. We used the same notation in Tables 1, 2, and 3 in the paper.

---

> > > > ### Comment · Reviewer_Cq8Z · 2025-06-09
> > > >
> > > > Thanks for the clarification.

---

> ### Author Response · Authors · 2025-06-02
> **Author Response (Part 2/4)**
>
> > Lack of context of this work relative to prior work on document expansion
>
> Thank you for pointing out the related work— we will ensure to include and compare it thoroughly in the related work section of the next version.
>
> Specifically, in contrast to document enrichment methods like doc2query, our approach expands enrichment across multiple modalities (e.g., documents and tables) and incorporates various enrichment types—including purpose, summary, and synthetic queries—while using off-the-shelf models rather than fine-tuned ones. Additionally, our method diverges in the online retrieval process: whereas doc2query concatenates generated queries with original documents for similarity search, we index each enrichment type separately and compute a weighted sum during retrieval. This design offers greater flexibility in storing vector embeddings, as each index operates independently and can be expanded with additional embeddings or indices as needed.
>
>
> > Unfair comparison with rerankers
>
> We used Llama-3.1-8B-Instruct as the online reranker because this was used in the original Judgerank paper and is a popular open-source model.
>
> We further use GPT-4o-mini as an alternative online re-ranker, the same model we employed for the offline enrichment process. The results, shown in the table below, indicate that switching from Llama-8B to GPT-4o-mini improves overall performance. Nevertheless, EnrichIndex still outperforms Judgerank even when the latter uses GPT-4o-mini. Furthermore, applying GPT-4o-mini for online re-ranking on top of EnrichIndex leads to even greater performance gains.
>
>
> R stands for recall @ 10, N stands for NDCG @ 10
>
> |                            | StackExchange (R) | StackExchange (N) | Coding (R) | Coding (N) | Theorems (R) | Theorems (N) | Average (R) | Average (N) |
> |-----------------------------------|--------------------|--------------------|-------------|-------------|----------------|----------------|---------------|---------------|
> | BM25                              | 37.7               | 32.8               | 12.2        | 9.5         | 13.7           | 9.8            | 26.4          | 22.2          |
> | Judgerank (BM25-J) with Llama8B   | 39.3               | 33.9               | 13.2        | 9.9         | 15.3           | 10.9           | 27.9          | 23.1          |
> | Judgerank with GPT-4o-mini        | 40.3               | 35.0               | 12.2        | 11.6        | 17.8           | 12.8           | 28.9          | 24.6          |
> | EnrichIndex                       | 39.7               | 34.7               | 20.2        | 17.0        | 20.7           | 14.4           | 30.9          | 25.8          |
> | EnrichIndex-J with Llama8B        | 40.0               | 35.5               | 20.3        | 17.8        | 21.2           | 14.7           | 31.2          | 26.5          |
> | EnrichIndex-J with GPT-4o-mini    | 40.2               | 35.5               | 22.8        | 21.2        | 22.5           | 16.5           | 32.1          | 27.6          |
>
>
> |                            | Spider2 (R) | Spider2 (N) | Beaver (R) | Beaver (N) | Fiben (R) | Fiben (N) | Average (R) | Average (N) |
> |-----------------------------------|-------------|-------------|-------------|-------------|------------|------------|--------------|--------------|
> | BM25                              | 39.1        | 28.6        | 47.0        | 37.5        | 46.2       | 34.5       | 44.0         | 33.3         |
> | Judgerank (BM25-J) with Llama8B   | 40.2        | 30.3        | 47.8        | 38.8        | 46.2       | 34.7       | 44.6         | 34.3         |
> | Judgerank with GPT-4o-mini        | 40.8        | 29.8        | 48.3        | 38.6        | 46.5       | 34.9       | 45.1         | 34.2         |
> | EnrichIndex                       | 66.4        | 51.7        | 62.1        | 55.8        | 65.7       | 51.5       | 65.0         | 52.7         |
> | EnrichIndex-J with Llama8B        | 66.9        | 52.6        | 61.8        | 55.7        | 65.6       | 51.7       | 65.0         | 53.1         |
> | EnrichIndex-J with GPT-4o-mini    | 67.7        | 52.8        | 60.8        | 54.9        | 66.0       | 51.9       | 65.2         | 53.0         |

---

> ### Author Response · Authors · 2025-06-02
> **Author Response (Part 1/4)**
>
> Thank you for your review. Please note that there appears to have been an error on the reviewer's side regarding the evaluation setting on BRIGHT (please see (*) below). In addition, we addressed the additional comments: adding new results of a stronger baseline, a thorough analysis of our weight tuning approach, and we will be sure to add an extended Related Work section containing all of the works referenced by the reviewers. Given the initial misunderstanding and our efforts to thoroughly address the concerns raised, we would sincerely appreciate it if the reviewer would consider revising their assessment and raising their score.
>
> (*) First of all, we would like to make two factual clarifications regarding the summary of the review.
>
> > The approach is evaluated on the coding part of the Bright dataset.
>
> > Finally reporting on a subset of Bright makes it appear that the dataset was cherry picked for the best subset of the technique. The remaining Bright numbers should at least appear in the appendix for the interested reader.
>
> As mentioned in Section 3.1 (line 139), Table 1, Table 4, and Table 5, we evaluated on the Bright dataset across **all three** top domains: StackExchange, Coding, and Theorems.
>
> > While demonstrating that the added information is about to move documents closer to the query using Wasserstein distance in Section 4.1 is somewhat interesting, this fails to capture the interaction between relevant and non-relevant information that is key to ranking relevant information above non-relevant information. It is not sufficient to move relevant information closer to the query if non-relevant information also moves closer to the query.
>
> Section 4.1 and Table 6 present the Wasserstein distance between the similarity distribution of (query, non-gold objects) **and** that of (query, gold objects), both before and after applying EnrichIndex—not just the distribution of (query, gold objects) alone. Our results show that EnrichIndex increases the distance between gold and non-gold similarity distributions, assisting retriever models in better distinguishing gold objects from non-gold ones. We believe this analysis reflects the interaction between relevant and irrelevant information with respect to the query, rather than considering only one side in isolation.

---

> > ### Comment · Reviewer_Cq8Z · 2025-06-07
> > **Reply 1**
> >
> > Thank you for explaining my misunderstanding in your work. For Bright, it would be helpful to at the performance on the individual subdomains to the appendix.

---

> > > ### Author Response · Authors · 2025-06-07
> > >
> > > Thank you! We will include the performance of each subdomain in the Appendix in the next version of the paper.

---

> > > > ### Comment · Reviewer_Cq8Z · 2025-06-09
> > > >
> > > > Thanks, I will look for those additional results in the appendix.

---

### Official Review · Reviewer_u3K1 · 2025-05-12

**Rating:** 7
**Confidence:** 5
**Ethics Flag:** 1

**Summary:**

This paper uses LLMs to augment retrieval objects (aka documents or chunks) with three new attributes: summaries, purpose, and potential queries (and answers). During retrieval, all attribute types (and the original object) are scored. This technique leads to strong results on challenging datasets.

**Questions To Authors:**

n/a

**Reasons To Accept:**

A1. It is becoming more clear recently that the top off-the-shelf retrieval systems are missing key capabilities. This paper presents a flexible way to address this shortcoming, and can be a foundational work for future research. I see it as quite different from typical Rank Fusion which acts more like an ensemble across models, as well as different from techniques that boost on specific metadata, e.g. popularity, quality, date, etc. Additionally, this is exactly the type of work that can bridge communities across IR and LLM research. That being said, it would be nice if it acknowledged more of the existing related research.

A2. There are strong results across a variety of interesting datasets. The results on BRIGHT are notable given the recent focus on agentic systems and finetuning with synthetic data, and potentially changes the narrative on what is required to do well on "reasoning" for IR. The results on Tables are particularly impressive with a huge improvement w/ BM25. And it's nice to have NQ as a sanity check.

A3. It's fascinating that different datasets benefited from different types of enriched info. This does make me wonder about generalization of this approach and how sensitive it will be to tuning hparams. It would have been nice to see more qualitative examples of queries that benefited from enrich differently.

A4. The paper includes a lot of detail to aid replication, including prompts.

**Reasons To Reject:**

R1. It was not clear to me whether it is important to use the exact similarity for each enriched info type. This could become not very practical if all similarities are required, and may be more prone to errors when using approximate nearest neighbors than retrieval that only uses one "type".

R2. Hyperparameter tuning. a. There is no analysis of hparam tuning. It is plausible that hparam tuning could be very important given the ablation shown in Fig 2. b. According to Appendix E, the results are reported only on 80% of the original test data with 20% of the test data being used for validation. It makes the results harder to compare against existing published results, e.g. for BRIGHT, but the overall findings remain interesting. c. AFAICT the hparams per dataset were not released. Why not?

R3. Too few cited work (<20) for such a highly researched topic, and not even a related work section. It is probably worth mentioning work that has focused on improving retrieval via summarization such as done in ALCE, RAPTOR, Decontextualization (Choi et al), among others. Also failures of neural IR have been well studied in papers like ABNIRML.

R4. JudgeRank is a complete strawman wrt cost. It is an agentic impractical method, and very few companies would consider deploying this---its main purpose is to show us that reranking on BRIGHT is possible, and perhaps can be distilled into a cheaper cross-encoder. I understand that it is current SOTA, hence is an appropriate target for comparison, but the focus on token counts as presented in the paper hides some of the real costs associated with using EnrichIndex.

Two hidden costs not mentioned in the paper:

a. The storage cost is 4x. If we assume 1k tokens per object at gpt-4o-mini output token pricing ($0.60/1M tokens) then EnrichIndex costs about $600 to enrich a 1M object corpus. In comparison, according to qdrant pricing, 1M vectors is about $70/month per replica, so +$210/month to support 4x vectors per object.

b. Similarly, if the enriched info is meant to be used in RAG, reranking, or other applications, then the input costs of those downstream pipelines will be increased.

---

> ### Author Response · Authors · 2025-06-02
> **Author Response (Part 3/3)**
>
> > R3: Too few cited work (<20) for such a highly researched topic, and not even a related work section. It is probably worth mentioning work that has focused on improving retrieval via summarization such as done in ALCE, RAPTOR, Decontextualization (Choi et al), among others. Also failures of neural IR have been well studied in papers like ABNIRML.
>
> Thank you for pointing out the related work— we will be sure to include all of the referenced works (ALCE, RAPTOR, Decontextualization (Choi et al) and ABNIRML) into a more comprehensive Related Work section.
>
> > R4: JudgeRank is a complete strawman wrt cost. It is an agentic impractical method, and very few companies would consider deploying this---its main purpose is to show us that reranking on BRIGHT is possible, and perhaps can be distilled into a cheaper cross-encoder. I understand that it is current SOTA, hence is an appropriate target for comparison, but the focus on token counts as presented in the paper hides some of the real costs associated with using EnrichIndex.
>
> We agree that Judgerank’s high computational cost can make it impractical for real-world deployment, which is precisely what motivated this work. We have shown that EnrichIndex not only outperforms Judgerank but can also operate at a significantly lower cost, as we explain below—making it a more practical and scalable solution for real-world applications.
>
> In terms of cost, our method primarily involves two components: offline enrichment and storage.
>
> > b. Similarly, if the enriched info is meant to be used in RAG, reranking, or other applications, then the input costs of those downstream pipelines will be increased.
>
> Importantly, during re-ranking, we only use the original object content, so there is no added cost during downstream tasks.
>
> > a. The storage cost is 4x. If we assume 1k tokens per object at gpt-4o-mini output token pricing (600 to enrich a 1M object corpus. In comparison, according to qdrant pricing, 1M vectors is about 210/month to support 4x vectors per object.
>
> For offline enrichment, it is worth noting that in real-world scenarios, corpora are often relatively static, meaning enrichment only needs to be performed **once**. When new data is added, enrichment can be applied incrementally to just those new objects. While this approach may increase storage requirements due to additional vector embeddings, the overall cost becomes negligible when amortized over a large number of queries.
>
> Additionally, latency is a critical factor in many applications. Our approach introduces only a minor increase in computation—specifically, a few extra similarity calculations—which adds minimal overhead compared to the latency introduced by LLM-based methods. This efficiency makes our method well-suited for high-throughput query environments.

---

> ### Author Response · Authors · 2025-06-02
> **Author Response (Part 2/3 continued)**
>
> R stands for recall @ 10, N stands for NDCG @ 10
>
> | 10% validation data (original query)                   | StackExchange (R) | StackExchange (N) | Coding (R) | Coding (N) | Theorems (R) | Theorems (N) | Average (R) | Average (N) |
> |--------------------------|-------------------|--------------------|------------|-------------|----------------|----------------|--------------|--------------|
> | Average                  | 22.6              | 18.7               | 16.9       | 12.0        | 9.3            | 6.1            | 17.8         | 14.0         |
> | Average with EnrichIndex | **23.5**              | **19.3**               | **18.5**       | **14.9**        | **10.8**           | **7.2**            | **19.1**         | **15.2**        |
>
>
> | 10% validation data (GPT-4 generated expanded query)                   | StackExchange (R) | StackExchange (N) | Coding (R) | Coding (N) | Theorems (R) | Theorems (N) | Average (R) | Average (N) |
> |--------------------------|-------------------|--------------------|------------|-------------|----------------|----------------|--------------|--------------|
> | Average                  | 33.0              | 29.0               | 15.1       | 11.9        | 16.3           | 11.7           | 25.1         | 21.1         |
> | Average with EnrichIndex | **33.2**              | 28.9               | **16.9**       | **15.1**        | **19.5**           | **13.6**          | **26.5**         | **22.1**         |
>
>
> | 10% validation data (original query)                   | Spider2 (R) | Spider2 (N) | Beaver (R) | Beaver (N) | Fiben (R) | Fiben (N) | Average (R) | Average (N) |
> |--------------------------|-------------|-------------|-------------|-------------|------------|------------|---------------|---------------|
> | Average                  | 53.9        | 41.5        | 53.1        | 45.7        | 47.3       | 42.9       | 51.1          | 43.2          |
> | Average with EnrichIndex | **64.3**        | **49.2**        | **59.3**        | **52.7**        | **52.2**       | **47.8**       | **58.2**          | **49.6**          |
>
> | 10% validation data (GPT-4 generated expanded query)                   | Spider2 (R) | Spider2 (N) | Beaver (R) | Beaver (N) | Fiben (R) | Fiben (N) | Average (R) | Average (N) |
> |--------------------------|-------------|-------------|-------------|-------------|------------|------------|---------------|---------------|
> | Average                  | 53.6        | 41.5        | 52.0        | 43.7        | 52.9       | 46.1       | 52.9          | 43.9          |
> | Average with EnrichIndex | **62.9**        | **49.6**        | **56.8**        | **50.9**        | **58.9**       | **50.3**       | **59.7**          | **50.2**          |

---

> > ### Author Response · Authors · 2025-06-09
> > **Author Response (Part 2/3 continued)**
> >
> > In addition, we analyzed whether a *single* set of coefficients could be tuned and applied across all datasets. To test this, we combined the validation sets from *all* datasets, Bright, Spider2, Beaver, and Fiben, into one unified validation set for the weight tuning process. The results, summarized in the table below, show that EnrichIndex still delivers substantial performance improvements, demonstrating the generalizability of our approach across diverse workloads simultaneously.
> >
> > R stands for recall @ 10, N stands for NDCG @ 10
> >
> > | Original query                   | StackExchange (R) | StackExchange (N) | Coding (R) | Coding (N) | Theorems (R) | Theorems (N) | Average (R) | Average (N) |
> > |--------------------------|-------------------|--------------------|------------|-------------|----------------|----------------|--------------|--------------|
> > | Average                  | 22.6              | 18.7               | 16.5       | 11.8        | 9.3            | 6.2            | 17.8         | 13.9         |
> > | Average with EnrichIndex | **24.1**              | **20.0**               | **17.8**       | **13.5**        | **10.8**           | **7.3**            | **19.3**         | **15.3**        |
> >
> >
> > | GPT-4 generated expanded query                   | StackExchange (R) | StackExchange (N) | Coding (R) | Coding (N) | Theorems (R) | Theorems (N) | Average (R) | Average (N) |
> > |--------------------------|-------------------|--------------------|------------|-------------|----------------|----------------|--------------|--------------|
> > | Average                  | 33.5              | 29.3               | 15.6       | 12.2        | 16.8           | 11.9           | 25.6         | 21.4         |
> > | Average with EnrichIndex | **34.7**              | **30.3**               | **17.2**       | **14.3**        | **19.5**           | **14.0**          | **27.3**         | **22.9**         |
> >
> >
> > | Original query                   | Spider2 (R) | Spider2 (N) | Beaver (R) | Beaver (N) | Fiben (R) | Fiben (N) | Average (R) | Average (N) |
> > |--------------------------|-------------|-------------|-------------|-------------|------------|------------|---------------|---------------|
> > | Average                  | 53.8        | 41.5        | 53.5        | 45.8        | 47.1       | 42.6       | 51.1          | 43.1         |
> > | Average with EnrichIndex | **63.7**        | **50.1**        | **58.2**        | **51.0**        | **52.2**       | **47.6**       | **57.7**          | **49.3**          |
> >
> > | GPT-4 generated expanded query                   | Spider2 (R) | Spider2 (N) | Beaver (R) | Beaver (N) | Fiben (R) | Fiben (N) | Average (R) | Average (N) |
> > |--------------------------|-------------|-------------|-------------|-------------|------------|------------|---------------|---------------|
> > | Average                  | 52.2        | 40.0        | 51.9        | 43.7        | 53.2       | 45.1       | 52.5          | 43.0          |
> > | Average with EnrichIndex | **62.8**        | **48.9**        | **56.7**        | **50.4**        | **57.6**       | **50.0**       | **59.1**          | **49.8**          |

---

> ### Author Response · Authors · 2025-06-02
> **Author Response (Part 2/3)**
>
> > R2. Hyperparameter tuning
>
> We provide a more thorough analysis of the hyperparameter tuning below. We will be sure to add this discussion to the final version of our paper.
>
> We note that this tuning process is highly parallelizable, requires minimal effort, and enables easy adaptation of our general retrieval framework to various workloads.
>
> > c. AFAICT the hparams per dataset were not released. Why not?
>
> The coefficients used in the weighted sums in Equation 1 are easy to reproduce, as they are determined by the simple process of enumerating all possible coefficient combinations that sum to one, followed by selecting the combination that yields the highest recall or NDCG@10 on a 20% validation set. We will provide the detailed procedure and the values in the next version of the paper.
>
> > a. There is no analysis of hparam tuning. It is plausible that hparam tuning could be very important given the ablation shown in Fig 2.
>
> To demonstrate the robustness and generalizability of our approach, we conducted a more thorough analysis of the coefficient tuning process. Namely, we evaluated whether a smaller portion of validation data could still yield strong performance. While the default setup uses 20% of the data for validation, we also experimented with using 15% and 10%, and reported the results for stage-one retrievers below. Overall, we observe that even with reduced validation data, EnrichIndex continues to deliver significant performance gains, highlighting the robustness of our tuning methodology.
>
> R stands for recall @ 10, N stands for NDCG @ 10
>
> | 15% validation data (original query)                   | StackExchange (R) | StackExchange (N) | Coding (R) | Coding (N) | Theorems (R) | Theorems (N) | Average (R) | Average (N) |
> |--------------------------|-------------------|--------------------|------------|-------------|----------------|----------------|--------------|--------------|
> | Average                  | 22.6              | 18.7               | 16.7       | 11.8        | 9.2            | 6.0            | 17.8         | 14.0         |
> | Average with EnrichIndex | **23.9**              | **19.4**               | **18.7**      | **14.6**       | **10.6**           | **7.2**            | **19.3**         | **15.2**         |
>
>
> | 15% validation data (GPT-4 generated expanded query)                   | StackExchange (R) | StackExchange (N) | Coding (R) | Coding (N) | Theorems (R) | Theorems (N) | Average (R) | Average (N) |
> |--------------------------|-------------------|--------------------|------------|-------------|----------------|----------------|--------------|--------------|
> | Average                  | 33.5              | 29.3               | 15.4       | 12.0        | 16.3           | 11.7           | 25.4         | 21.2         |
> | Average with EnrichIndex | **33.7**              | **29.4**               | **17.4**       | **15.5**        | **19.6**           | **13.8**           | **26.8**         | **22.5**        |
>
>
> | 15% validation data (original query)                   | Spider2 (R) | Spider2 (N) | Beaver (R) | Beaver (N) | Fiben (R) | Fiben (N) | Average (R) | Average (N) |
> | ------------------------ | ----------- | ----------- | ---------- | ---------- | --------- | --------- | ----------- | ----------- |
> | Average                  | 54.0        | 41.9        | 53.1       | 45.7       | 47.3      | 42.9      | 51.1        | 43.3        |
> | Average with EnrichIndex | **64.5**        | **49.9**        | **58.6**       | **52.4**       | **54.1**      | **49.3**      | **58.8**        | **50.3**        |
>
>
> | 15% validation data (GPT-4 generated expanded query)                   | Spider2 (R) | Spider2 (N) | Beaver (R) | Beaver (N) | Fiben (R) | Fiben (N) | Average (R) | Average (N) |
> | ------------------------ | ----------- | ----------- | ---------- | ---------- | --------- | --------- | ----------- | ----------- |
> | Average                  | 52.8        | 40.9        | 52.0       | 43.8       | 53.6      | 46.0      | 52.9        | 43.7        |
> | Average with EnrichIndex | **63.1**        | **49.4**        | **58.5**       | **52.1**       | **60.0**      | **51.6**      | **60.6**       | **51.0**        |

---

> ### Author Response · Authors · 2025-06-02
> **Author Response (Part 1/3)**
>
> We appreciate the highly positive review, noting that EnrichIndex may serve as a foundation for future research bridging IR and LLM research. We will be sure to add a more comprehensive overview of the Related Work to our manuscript. Regarding the reviewer’s comments:
>
> >  R1: It was not clear to me whether it is important to use the exact similarity for each enriched info type. This could become not very practical if all similarities are required, and may be more prone to errors when using approximate nearest neighbors than retrieval that only uses one "type".
>
> The core objective of the online retrieval phase is to identify the top-k objects based on multiple attributes. In the paper, we present a simple approach: computing the weighted sum of similarities for all objects and selecting the top-k results. However, other alternatives exist for retrieving the optimal object based on a series of rankings: for example, by combining nearest neighbor search with the classical Fagin’s algorithm (or the Threshold Algorithm). Fagin’s algorithm is designed to efficiently retrieve the top-k items from multiple sorted lists when using a monotonic aggregation function—such as the one defined in Equation 1. The partially sorted lists can be obtained by performing top-k nearest neighbor searches based on query-object similarities for each enrichment type.

---

### Official Review · Reviewer_LPFq · 2025-05-12

**Rating:** 6
**Confidence:** 4
**Ethics Flag:** 1

**Summary:**

This paper aims to solve a practical latency problem when using LLMs for ranking documents or tables in information retrieval.  The basic idea is to extract semantic information from objects and then use such information as additional representation of the objects. In this paper, three specific representations are considered, which are summary, purpose, and QA pairs. Experimental results show that the proposed approach improve stage-one retriever a lot, and the approach also outperforms LLM ranking system such as JudgeRank.

**Questions To Authors:**

Authors compare JudgeRank and the proposed approach in order to demonstrate the effectiveness of the proposed approach. But, authors haven't told what LlM is used  in the implementation of JudgeRank. Does it also use GPT 4o-mini?  If it uses other models, such as LLaMa models, the performance comparison will be unfair.

**Reasons To Accept:**

This paper discusses a practical issue that practitioners will meet  when using LLMs for ranking documents or tables. Such research has practical value. In addition, experimental results also demonstrate the effectiveness of the proposed approach.

**Reasons To Reject:**

1. The proposed approach is straightforward. Authors are suggested to add more discussion on why summary, purpose, and QA pairs  help to improve performance in this task.
2. In my opinion, the section of "Performance of stage-one retrievers with query expansion" is not very necessary.

---

> ### Author Response · Authors · 2025-06-02
> **Author Response (Part 1/1)**
>
> Thank you for your review and feedback. We hope our response below bolsters the soundness of our method and effectively addresses the concerns about our paper.
>
> > The proposed approach is straightforward. Authors are suggested to add more discussion on why summary, purpose, and QA pairs help to improve performance in this task.
>
> Following the reviewer's suggestions, we intend to add an expanded Related Work to the paper, where we will also discuss our enrichment types (summary, purpose, QA) and their relation with prior work.
>
>
> **Questions**
>
> > Authors compare JudgeRank and the proposed approach in order to demonstrate the effectiveness of the proposed approach. But, authors haven't told what LlM is used in the implementation of JudgeRank. Does it also use GPT 4o-mini? If it uses other models, such as LLaMa models, the performance comparison will be unfair.
>
> The LLM used by the JudgeRank baseline is Llama-3.1-8B-Instruct, also mentioned in Table 4. This is due to Llama-8B being used in the original Judgerank paper, which was the state-of-the-art on the BRIGHT benchmark.
>
> Following the reviewer’s suggestion, we ran new experiments with a stronger version of JudgeRank using GPT-4o-mini. The results, in the table below, show that EnrichIndex continues to outperform JudgeRank, even when the baseline is powered by GPT-4o-mini. Furthermore, applying GPT-4o-mini for online re-ranking on top of EnrichIndex leads to even greater performance gains over JudgeRank.
>
>
> R stands for recall @ 10, N stands for NDCG @ 10
>
> |                            | StackExchange (R) | StackExchange (N) | Coding (R) | Coding (N) | Theorems (R) | Theorems (N) | Average (R) | Average (N) |
> |-----------------------------------|--------------------|--------------------|-------------|-------------|----------------|----------------|---------------|---------------|
> | BM25                              | 37.7               | 32.8               | 12.2        | 9.5         | 13.7           | 9.8            | 26.4          | 22.2          |
> | Judgerank (BM25-J) with Llama8B   | 39.3               | 33.9               | 13.2        | 9.9         | 15.3           | 10.9           | 27.9          | 23.1          |
> | Judgerank with GPT-4o-mini        | 40.3               | 35.0               | 12.2        | 11.6        | 17.8           | 12.8           | 28.9          | 24.6          |
> | EnrichIndex                       | 39.7               | 34.7               | 20.2        | 17.0        | 20.7           | 14.4           | 30.9          | 25.8          |
> | EnrichIndex-J with Llama8B        | 40.0               | 35.5               | 20.3        | 17.8        | 21.2           | 14.7           | 31.2          | 26.5          |
> | EnrichIndex-J with GPT-4o-mini    | 40.2               | 35.5               | 22.8        | 21.2        | 22.5           | 16.5           | 32.1          | 27.6          |
>
>
> |                            | Spider2 (R) | Spider2 (N) | Beaver (R) | Beaver (N) | Fiben (R) | Fiben (N) | Average (R) | Average (N) |
> |-----------------------------------|-------------|-------------|-------------|-------------|------------|------------|--------------|--------------|
> | BM25                              | 39.1        | 28.6        | 47.0        | 37.5        | 46.2       | 34.5       | 44.0         | 33.3         |
> | Judgerank (BM25-J) with Llama8B   | 40.2        | 30.3        | 47.8        | 38.8        | 46.2       | 34.7       | 44.6         | 34.3         |
> | Judgerank with GPT-4o-mini        | 40.8        | 29.8        | 48.3        | 38.6        | 46.5       | 34.9       | 45.1         | 34.2         |
> | EnrichIndex                       | 66.4        | 51.7        | 62.1        | 55.8        | 65.7       | 51.5       | 65.0         | 52.7         |
> | EnrichIndex-J with Llama8B        | 66.9        | 52.6        | 61.8        | 55.7        | 65.6       | 51.7       | 65.0         | 53.1         |
> | EnrichIndex-J with GPT-4o-mini    | 67.7        | 52.8        | 60.8        | 54.9        | 66.0       | 51.9       | 65.2         | 53.0         |

---

> > ### Comment · Reviewer_LPFq · 2025-06-10
> >
> > Thanks for providing additional experiments to answer my questions. I think the response is quite acceptable.

---

### Official Review · Reviewer_DMiz · 2025-05-12

**Rating:** 7
**Confidence:** 4
**Ethics Flag:** 1

**Summary:**

This paper introduces EnrichIndex, a novel data augmentation technique that enhances information retrieval by using Large Language Models (LLMs) offline. During data ingestion, EnrichIndex performs a single pass over the corpus, generating summaries, purpose statements, and question-answer pairs for each document or table. Unlike online LLM approaches that process each query, EnrichIndex utilizes these pre-computed enrichments alongside the original content during retrieval, calculating a weighted similarity score. Evaluations on passage and table retrieval show consistent improvements over baselines. Crucially, this offline approach leads to substantially lower online latency and cost compared to online LLM re-ranking methods.

**Questions To Authors:**

- Could you please report statistics on the generated enrichments, such as average length per type (summary, purpose) and the number of QA pairs per item.
- Can you share qualitative analysis (e.g., case studies, examples) demonstrating how the enrichments specifically improve table retrieval.
- What are the potential biases (e.g., factual errors, skewed perspectives from the LLM) or limitations (e.g., generation of overly verbose or irrelevant content) that might be introduced by the LLM-generated enrichments, and what methodologies could be employed to mitigate such concerns?

**Reasons To Accept:**

- EnrichIndex offers a practical and efficient solution by utilizing LLMs offline for index enrichment, leading to consistent performance gains.
- The evaluation is comprehensive, covering diverse retrieval tasks and datasets. The results demonstrate clear improvements in both retrieval performance and efficiency, particularly noticeable on text-scarce table retrieval benchmarks.
- The paper provides a detailed analysis of the contribution of each enrichment type (summary, purpose, QA pairs), offering valuable insights.

**Reasons To Reject:**

- The current submission lacks a discussion of related studies. Specifically, it needs to acknowledge and contextualize prior work on data augmentation techniques for boosting retrieval performance, including but not limited to works such as [1-5]. Given its direct relevance to pseudo query augmentation (QA pairs), a direct comparison to [1] is particularly encouraged.
- The models and baselines employed in the experimental evaluation appear to be dated. To provide a more compelling demonstration of  EnrichIndex's advantages, comparison against more current state-of-the-art retrieval methods is recommended.
- While the paper mentions the tuning of weights for combining similarity scores, crucial details regarding the methodology are absent.
- The exclusive reliance on GPT-4o-mini for data enrichment and expansion raises questions about the generalizability and replicability of the reported results across different Large Language Models.


[1] Wang, Zhengren, et al. "QAEncoder: Towards Aligned Representation Learning in Question Answering System." arXiv preprint arXiv:2409.20434 (2024).
[2] Tan, Hongming, et al. "QAEA-DR: a unified text augmentation framework for dense retrieval." IEEE Transactions on Knowledge and Data Engineering (2025).
[3] Gao, Luyu, et al. "Precise zero-shot dense retrieval without relevance labels." Proceedings of the 61st Annual Meeting of the Association for Computational Linguistics (Volume 1: Long Papers). 2023.
[4] Shen, Tao, et al. "Retrieval-augmented retrieval: Large language models are strong zero-shot retriever." Findings of the Association for Computational Linguistics ACL 2024. 2024.
[5] Sun, Dong, et al. "Zero-shot Document Retrieval with Hybrid Pseudo-document Retriever." ICASSP 2025-2025 IEEE International Conference on Acoustics, Speech and Signal Processing (ICASSP). IEEE, 2025.

---

> ### Author Response · Authors · 2025-06-02
> **Author Response (Part 4/4)**
>
> **Questions**
>
> > Could you please report statistics on the generated enrichments, such as average length per type (summary, purpose) and the number of QA pairs per item.
>
> We provide the enrichment statistics (with the GPT-4o-mini model), using OpenAI's tokenizer: we found that the average token count is 65.1 for enriched summaries, 76.5 for enriched purposes, and 477.7 for QA pairs — with an average of 16.5 QA pairs per object. Overall, the token count for enriching each object is small. In addition, note that the number of QA pairs can be easily adjusted by specifying the desired amount directly in the prompt.
>
> > Can you share qualitative analysis (e.g., case studies, examples) demonstrating how the enrichments specifically improve table retrieval.
>
> In the original paper, we included two examples—Figure 1 for tables and Figure 3 in the appendix for documents—to illustrate how our enrichment process works and how it enhances retrieval effectiveness.
>
> > What are the potential biases (e.g., factual errors, skewed perspectives from the LLM) or limitations (e.g., generation of overly verbose or irrelevant content) that might be introduced by the LLM-generated enrichments, and what methodologies could be employed to mitigate such concerns?
>
> As with all LLM-generated text, there is always the risk of enrichments that include hallucinated data. It would definitely be interesting to incorporate fact verification techniques together with EnrichIndex to better counter hallucination during the offline process. While this serves as an exciting topic for future work, we believe it is orthogonal to our current paper.

---

> ### Author Response · Authors · 2025-06-02
> **Author Response (Part 3/4)**
>
> > The exclusive reliance on GPT-4o-mini for data enrichment and expansion raises questions about the generalizability and replicability of the reported results across different Large Language Models.
>
> We selected GPT-4o-mini for offline enrichment because it is a strong and widely used proprietary model that offers a good balance between performance and cost. However, to address the reviewer’s concerns, we also added results using the popular open-source Llama-3.1-8B-Instruct model as an alternative for offline enrichment. The results below show that even when using enrichment generated by Llama-8B, EnrichIndex consistently yields significant improvements across different stage-one retrievers.
>
> Due to limited computational resources, we applied this model to all table datasets and one representative sub-domain from each major category in the Bright benchmark. The table below presents the average retrieval performance of various stage-one retrievers, both with and without EnrichIndex.
>
> R stands for recall @ 10, N stands for NDCG @ 10
>
> | Original query                  | StackExchange-Biology (R) | StackExchange-Biology (N) | Coding-Pony (R) | Coding-Pony (N) | Theorems-TheoremQA_T (R) | Theorems-TheoremQA_T (N) | Average (R) | Average (N) |
> |-------------------------|--------------------------------------|------------------------------------|----------------------------|--------------------------|------------------------------------|----------------------------------|--------------------------|------------------------|
> | Average                 | 20.5                                 | 18.2                               | 4.1                        | 6.7                      | 8.0                                | 4.9                              | 11.0                     | 10.3                   |
> | Average with EnrichIndex| **26.3**                                 | **22.2**                               | **7.3**                        | **12.7**                     | **12.6**                               | **8.4**                              | **15.4**                     | **15.0**                   |
>
> | GPT-4 generated expanded query                  | StackExchange-Biology (R) | StackExchange-Biology (N) | Coding-Pony (R) | Coding-Pony (N) | Theorems-TheoremQA_T (R) | Theorems-TheoremQA_T (N) | Average (R) | Average (N) |
> |-------------------------|--------------------------------------|------------------------------------|----------------------------|--------------------------|------------------------------------|----------------------------------|--------------------------|------------------------|
> | Average                 | 39.8                                 | 34.7                               | 4.9                        | 8.2                      | 31.2                               | 24.0                             | 24.1                     | 21.7                   |
> | Average with EnrichIndex| **44.0**                                 | **38.8**                               | **8.2**                        | **15.4**                     | **32.3**                               | **25.3**                             | **27.2**                     | **26.3**                   |
>
>
> | Original query                   | Spider2 (R) | Spider2 (N) | Beaver (R) | Beaver (N) | Fiben (R) | Fiben (N) | Average (R) | Average (N) |
> |--------------------------|------------------------|----------------------|------------------------|----------------------|----------------------|--------------------|--------------------------|------------------------|
> | Average                  | 54.0                   | 42.0                 | 53.1                   | 45.7                 | 47.4                 | 42.8               | 51.2                     | 43.3                   |
> | Average with EnrichIndex | **59.4**                   | **45.9**                 | **58.9**                   | **51.4**                 | **59.0**                 | **56.5**               | **59.1**                     | **51.5**                   |
>
> | GPT-4 generated expanded query                   | Spider2 (R) | Spider2 (N) | Beaver (R) | Beaver (N) | Fiben (R) | Fiben (N) | Average (R) | Average (N) |
> |--------------------------|------------------------|----------------------|------------------------|----------------------|----------------------|--------------------|--------------------------|------------------------|
> | Average                  | 53.8                   | 41.5                 | 51.9                   | 43.9                 | 53.4                 | 45.6               | 53.1                     | 43.7                   |
> | Average with EnrichIndex | **59.1**                   | **46.2**                 | **54.1**                   | **46.9**                 | **63.3**                 | **57.1**               | **59.4**                     | **50.7**                   |

---

> ### Author Response · Authors · 2025-06-02
> **Author Response (Part 2/4)**
>
> > The models and baselines employed in the experimental evaluation appear to be dated. To provide a more compelling demonstration of EnrichIndex's advantages, comparison against more current state-of-the-art retrieval methods is recommended.
>
> Our baselines for implicit retrieval and table retrieval were considered to be state-of-the-art when we submitted our paper (late March 2025). The dense retrieval models that were used were among the top retrievers on the MTEB leaderboard, while JudgeRank, the LLM-based re-ranker, was the state-of-the-art on the Bright leaderboard. We will be sure to emphasize this further in our revised manuscript.
>
> In addition, we added new results, using a stronger version of JudgeRank — switching the underlying Llama-3.1-8B with the stronger GPT-4o-mini. Our results (in the table below) still demonstrate that EnrichIndex outperforms the GPT-4o-mini powered JudgeRank. Furthermore, combining GPT-4o-mini with EnrichIndex (in the online re-ranking phase) leads to an even greater performance gain compared to the GPT-4o-mini-powered JudgeRank. We will be sure to include these results, using a stronger baseline, in the revised version of our paper.
>
> R stands for recall @ 10, N stands for NDCG @ 10
>
> |                            | StackExchange (R) | StackExchange (N) | Coding (R) | Coding (N) | Theorems (R) | Theorems (N) | Average (R) | Average (N) |
> |-----------------------------------|--------------------|--------------------|-------------|-------------|----------------|----------------|---------------|---------------|
> | BM25                              | 37.7               | 32.8               | 12.2        | 9.5         | 13.7           | 9.8            | 26.4          | 22.2          |
> | Judgerank (BM25-J) with Llama8B   | 39.3               | 33.9               | 13.2        | 9.9         | 15.3           | 10.9           | 27.9          | 23.1          |
> | Judgerank with GPT-4o-mini        | 40.3               | 35.0               | 12.2        | 11.6        | 17.8           | 12.8           | 28.9          | 24.6          |
> | EnrichIndex                       | 39.7               | 34.7               | 20.2        | 17.0        | 20.7           | 14.4           | 30.9          | 25.8          |
> | EnrichIndex-J with Llama8B        | 40.0               | 35.5               | 20.3        | 17.8        | 21.2           | 14.7           | 31.2          | 26.5          |
> | EnrichIndex-J with GPT-4o-mini    | 40.2               | 35.5               | 22.8        | 21.2        | 22.5           | 16.5           | 32.1          | 27.6          |
>
>
> |                            | Spider2 (R) | Spider2 (N) | Beaver (R) | Beaver (N) | Fiben (R) | Fiben (N) | Average (R) | Average (N) |
> |-----------------------------------|-------------|-------------|-------------|-------------|------------|------------|--------------|--------------|
> | BM25                              | 39.1        | 28.6        | 47.0        | 37.5        | 46.2       | 34.5       | 44.0         | 33.3         |
> | Judgerank (BM25-J) with Llama8B   | 40.2        | 30.3        | 47.8        | 38.8        | 46.2       | 34.7       | 44.6         | 34.3         |
> | Judgerank with GPT-4o-mini        | 40.8        | 29.8        | 48.3        | 38.6        | 46.5       | 34.9       | 45.1         | 34.2         |
> | EnrichIndex                       | 66.4        | 51.7        | 62.1        | 55.8        | 65.7       | 51.5       | 65.0         | 52.7         |
> | EnrichIndex-J with Llama8B        | 66.9        | 52.6        | 61.8        | 55.7        | 65.6       | 51.7       | 65.0         | 53.1         |
> | EnrichIndex-J with GPT-4o-mini    | 67.7        | 52.8        | 60.8        | 54.9        | 66.0       | 51.9       | 65.2         | 53.0         |
>
>
> > While the paper mentions the tuning of weights for combining similarity scores, crucial details regarding the methodology are absent.
>
> As mentioned in Appendix E, for each dataset, we randomly split the questions into an 80/20 ratio, using 20% for tuning the weights. Overall, in order to optimize the four weights in Equation 1, we allowed each to range from 0.0 to 1.0 in increments of 0.1, ensuring their sum equals 1. This resulted in 286 possible combinations (calculated using the stars and bars method). We then parallelized the search across these combinations to identify the one that maximizes recall or NDCG@10 on the validation set. This tuning process is very efficient and requires minimal effort, while also allowing our general retrieval framework to be easily adapted to different workloads.  We would be happy to move some of the implementation details to the main paper in the camera-ready version.

---

> > ### Comment · Reviewer_DMiz · 2025-06-08
> >
> > Thanks for the response. But it is still not clear what the weights α1, α2, α3, α4 are. Can you share the numbers and ablation studies? Do you tune different weights for each dataset? It's important for us to understand the contribution of each enrichment.

---

> > > ### Author Response · Authors · 2025-06-09
> > >
> > > In the paper, we tuned the weights for each dataset. The tuning process mentioned above is easily reproducible, highly parallelizable, requires minimal effort, and enables easy adaptation of our general retrieval framework to various workloads.
> > >
> > > The weights for each dataset are as follows, in the order of (original content, purpose, summary, QA pairs):
> > > - Bright: (0.18, 0.29, 0.25, 0.28)
> > > - Beaver: (0.33, 0.47, 0.1, 0.1)
> > > - Spider2: (0.23, 0.47, 0.23, 0.07)
> > > - Fiben: (0.37, 0.0, 0.4, 0.23)
> > >
> > > Please note that the paper also includes ablation studies that evaluate performance across various datasets using different combinations of enrichment. As illustrated in Figure 2, for the Bright and Beaver datasets, the “purpose” enrichment yields the highest performance gain, followed by “QA pairs” and then “summary” enrichments. On the Spider2 dataset, “purpose” again leads, followed by the “summary” and “QA pairs” enrichments. On the Fiben dataset, “summary” has the greatest impact, followed by “QA pairs” and then “purpose”. These patterns are consistent with the relative magnitudes of the weights, indicating that higher weights generally correspond to greater contributions.
> > >
> > > In addition, we explored whether a *single* set of coefficients could be tuned and applied across all datasets. To test this, we combined the validation sets from *all* datasets, Bright, Spider2, Beaver, and Fiben, into one unified validation set for the weight tuning process. The results, summarized in the table below, show that EnrichIndex still delivers substantial performance improvements, demonstrating the generalizability of our approach across diverse workloads simultaneously.
> > >
> > > The weights are (0.4, 0.2, 0.2, 0.2) in the order of (original content, purpose, summary, QA pairs). Using this single set of weights, each type of enrichment appears to contribute fairly evenly.
> > >
> > > R stands for recall @ 10, N stands for NDCG @ 10
> > >
> > > | Original query                   | StackExchange (R) | StackExchange (N) | Coding (R) | Coding (N) | Theorems (R) | Theorems (N) | Average (R) | Average (N) |
> > > |--------------------------|-------------------|--------------------|------------|-------------|----------------|----------------|--------------|--------------|
> > > | Average                  | 22.6              | 18.7               | 16.5       | 11.8        | 9.3            | 6.2            | 17.8         | 13.9         |
> > > | Average with EnrichIndex | **24.1**              | **20.0**               | **17.8**       | **13.5**        | **10.8**           | **7.3**            | **19.3**         | **15.3**        |
> > >
> > >
> > > | GPT-4 generated expanded query                   | StackExchange (R) | StackExchange (N) | Coding (R) | Coding (N) | Theorems (R) | Theorems (N) | Average (R) | Average (N) |
> > > |--------------------------|-------------------|--------------------|------------|-------------|----------------|----------------|--------------|--------------|
> > > | Average                  | 33.5              | 29.3               | 15.6       | 12.2        | 16.8           | 11.9           | 25.6         | 21.4         |
> > > | Average with EnrichIndex | **34.7**              | **30.3**               | **17.2**       | **14.3**        | **19.5**           | **14.0**          | **27.3**         | **22.9**         |
> > >
> > >
> > > | Original query                   | Spider2 (R) | Spider2 (N) | Beaver (R) | Beaver (N) | Fiben (R) | Fiben (N) | Average (R) | Average (N) |
> > > |--------------------------|-------------|-------------|-------------|-------------|------------|------------|---------------|---------------|
> > > | Average                  | 53.8        | 41.5        | 53.5        | 45.8        | 47.1       | 42.6       | 51.1          | 43.1         |
> > > | Average with EnrichIndex | **63.7**        | **50.1**        | **58.2**        | **51.0**        | **52.2**       | **47.6**       | **57.7**          | **49.3**          |
> > >
> > > | GPT-4 generated expanded query                   | Spider2 (R) | Spider2 (N) | Beaver (R) | Beaver (N) | Fiben (R) | Fiben (N) | Average (R) | Average (N) |
> > > |--------------------------|-------------|-------------|-------------|-------------|------------|------------|---------------|---------------|
> > > | Average                  | 52.2        | 40.0        | 51.9        | 43.7        | 53.2       | 45.1       | 52.5          | 43.0          |
> > > | Average with EnrichIndex | **62.8**        | **48.9**        | **56.7**        | **50.4**        | **57.6**       | **50.0**       | **59.1**          | **49.8**          |

---

> > > > ### Comment · Reviewer_DMiz · 2025-06-09
> > > >
> > > > Thanks for providing additional information. The results from the single set of coefficients are compelling.
> > > >
> > > > To help readers better understand the contributions, I suggest updating the paper to clearly compare the performance across the different settings: the baseline, the single-set coefficient model, and the individually-tuned model.
> > > >
> > > > I will keep my rating unchanged at this time, as I believe the study would benefit from including other relevant baselines for a more comprehensive comparison.

---

> ### Author Response · Authors · 2025-06-02
> **Author Response (Part 1/4)**
>
> Thank you for your review, we appreciate the positive feedback. We hope to address the comments regarding the related work and analysis of our results below:
>
> > The current submission lacks a discussion of related studies. Specifically, it needs to acknowledge and contextualize prior work on data augmentation techniques for boosting retrieval performance, including but not limited to works such as [1-5]. Given its direct relevance to pseudo query augmentation (QA pairs), a direct comparison to [1] is particularly encouraged.
>
> We will be sure to add a comprehensive Related Work to our paper and to include and discuss all papers mentioned by the reviewer.

---

### Decision · Program_Chairs · 2025-07-08

**Decision:**

Accept

**Comment:**

The paper introduces an offline enrichment pipeline that augments each document with LLM-generated summaries, purpose statements, and Q&A pairs, then blends these artifacts with the original content at retrieval time. Reviewers appreciated the simplicity of this design, its low online latency, and the consistent improvements it yielded on BRIGHT, Natural-Questions, and several table-retrieval benchmarks---even when the enrichment was produced by a smaller open-weights model rather than GPT-4o.

Discussion focused on four issues: (i) storage overhead, (ii) fairness of the JudgeRank comparison, (iii) details of setting of the weighting hyperparams, and (iv) an initially sparse treatment of related work in document expansion. The authors responded with new JudgeRank experiments using comparable model sizes, further details on weight selection and generalization across datasets, and a commitment to expand the literature review. Reviewers indicated that these clarifications addressed their principal concerns.

Given the generally favorable assessments and the satisfactory responses, I believe the work is in good shape. I suggest integrating the cost and single-weight analyses into the main paper, expanding the related-work section to cover recent document-expansion approaches, and including more qualitative examples that illustrates how the enriched artifacts surface results that the base index alone would miss.